# The transferability of lipid loci across African, Asian and European cohorts

Karoline Kuchenbaecker [1,2,3], Nikita Telkar[4], Theresa Reiker[3,5,6,7], Robin G. Walters [8,9], Kuang Lin[9], Anders Eriksson[10], Deepti Gurdasani [3], Arthur Gilly[3,11], Lorraine Southam [3,11,12], Emmanouil Tsafantakis[13], Maria Karaleftheri[14], Janet Seeley [15,16,17], Anatoli Kamali[17], Gershim Asiki[17,18,19], Iona Y. Millwood[8,9], Michael Holmes[8,9], Huaidong Du[8,9], Yu Guo[20], Understanding Society Scientific Group, Meena Kumari[21], George Dedoussis[22], Liming Li[23], Zhengming Chen[9], Manjinder S. Sandhu [24] & Eleftheria Zeggini[3,11]

Most genome-wide association studies are based on samples of European descent. We assess whether the genetic determinants of blood lipids, a major cardiovascular risk factor, are shared across populations. Genetic correlations for lipids between European-ancestry and Asian cohorts are not significantly different from 1. A genetic risk score based on LDL-cholesterol-associated loci has consistent effects on serum levels in samples from the UK, Uganda and Greece ($r = 0.23$–$0.28$, $p < 1.9 \times 10^{-14}$). Overall, there is evidence of reproducibility for ~75% of the major lipid loci from European discovery studies, except triglyceride loci in the Ugandan samples (10% of loci). Individual transferable loci are identified using trans-ethnic colocalization. Ten of fourteen loci not transferable to the Ugandan population have pleiotropic associations with BMI in Europeans; none of the transferable loci do. The non-transferable loci might affect lipids by modifying food intake in environments rich in certain nutrients, which suggests a potential role for gene-environment interactions.

[1] Division of Psychiatry, University College of London, London W1T 7NF, UK. [2] UCL Genetics Institute, University College London, London WC1E 6BT, UK. [3] Department of Human Genetics, Wellcome Sanger Institute, Hinxton CB10 1SA, UK. [4] Department of Genetics, Evolution and Environment, University College London, London WC1E 6BT, UK. [5] Department of Public Health and Primary Care, University of Cambridge, Cambridge CB1 8RN, UK. [6] Swiss Tropical and Public Health Institute, Basel, Switzerland. [7] University of Basel, Basel, Switzerland. [8] Medical Research Council Population Health Research Unit (MRC PHRU), Nuffield Department of Population Health, University of Oxford, Oxford, UK. [9] Clinical Trial Service Unit and Epidemiological Studies Unit (CTSU), Nuffield Department of Population Health, University of Oxford, Oxford, UK. [10] Department of Medical and Molecular Genetics, King's College London, London SE1 9RT, UK. [11] Institute of Translational Genomics, Helmholtz Zentrum München, German Research Center for Environmental Health, Neuherberg, Germany. [12] Wellcome Trust Centre for Human Genetics, University of Oxford, Oxford OX3 7BN, UK. [13] Anogia Medical Centre, Anogia 740 51, Greece. [14] Echinos Medical Centre, Echinos, Xanthi 67300, Greece. [15] Department of Global Health and Development, London School of Hygiene & Tropical Medicine, London WC1E 7HT, UK. [16] Faculty of Public Health and Policy, London School of Hygiene & Tropical Medicine, London WC1E 7HT, UK. [17] Medical Research1 Council/Uganda Virus Research Institute and London School of Hygiene (MRC/UVRI and LSHTM), Uganda Research Unit, Entebbe, Uganda. [18] African Population and Health Research Center, Nairobi, Kenya. [19] Department of Women's and Children's Health, Karolinska Institutet, Stockholm, Sweden. [20] Chinese Academy of Medical Sciences, Beijing 100730, China. [21] Institute for Social and Economic Research, University of Essex, Wivenhoe Park, Colchester, Essex, UK. [22] Department of Nutrition and Dietetics, School of Health Science and Education, Harokopio University of Athens, Athens, Greece. [23] Department of Epidemiology and Biostatistics, School of Public Health, Peking University, Beijing 100191, China. [24] Department of Medicine, University of Cambridge, Cambridge CB2 0QQ, UK. Members of the Understanding Society Scientific Group are listed at the end of this paper. Correspondence and requests for materials should be addressed to K.K. (email: k.kuchenbaecker@ucl.ac.uk)

Genome-wide association studies (GWAS) have been very successful in identifying genetic variants linked to cardiovascular disease (CVD) and to cardiometabolic traits[1]. Due to the improving predictive accuracy of these variants, genetic risk prediction could soon be implemented in clinical settings[2,3]. However, the majority of samples included in these genome "white" association studies were British or US-Americans with European ancestry[4,5] which does not accurately represent the ethnically and ancestrally diverse populations of these nations. Moreover, three quarters of CVD-associated deaths occur in low- and middle-income countries where incidences are rising[6]. Consequently, it is important to determine whether cardiometabolic loci are transferable to other populations.

Previous research assessed the effects of different allele frequencies and linkage disequilibrium (LD) on genetic associations across ancestry groups[7]. Here we ask the fundamental question whether causal variants for blood lipids, a major cardiovascular risk factor, are shared across populations. Heterogeneity in effects of variants could result from epistasis or gene-environment interactions. However, the causal variants are usually unknown. The differences in LD structure between populations make it difficult to compare the observed associations between ancestry groups because the effect of a variant depends on its correlation with the causal variant(s)[7]. Differences in allele frequency also impact the power to detect associations in other ancestry groups.

We employ several strategies which account for these effects and do not require knowledge of the specific causal variants to quantify the extent to which genetic variants affecting lipid biomarkers are shared between individuals from Europe/North America, Asia, and Africa. We assess the transferability of individual signals and compare association patterns across the genome using data from the African Partnership for Chronic Disease Research – Uganda (APCDR-Uganda, $N = 6407$)[8], China Kadoorie Biobank (CKB, $N = 21,295$)[9], the Hellenic Isolated Cohorts (HELIC-MANOLIS, $N = 1641$ and HELIC-Pomak, $N = 1945$)[10,11], and the UK Household Longitudinal Study (UKHLS, $N = 9961$)[12]. We also use summary statistics from Biobank Japan (BBJ, $N = 162,255$)[13] and the Global Lipid Genetics Consortium (European ancestry, GLGC2013 $N = 188,577$, GLGC2017 $N = 237,050$)[14,15]. We find evidence for extensive sharing of genetic variants that affect levels of HDL- and LDL-cholesterol and triglycerides between individuals with European ancestry and samples from China, Japan and Greek population isolates. We estimate that about three quarters of major lipid loci are reproducible. Using trans-ethnic colocalization, we show that many established loci for triglycerides do not affect levels of this biomarker in Ugandan samples, however. Ten out of fourteen of the lipid loci that were not transferable to the Ugandan samples had pleiotropic associations with BMI in European ancestry samples. None of the transferable loci were linked to BMI. This could point to a role of environmental factors in modifying which genetic variants affect lipid levels.

## Results

### Reproducibility of established lipid loci.
We assessed rates at which established lipid-associated variants were reproducible in other populations. We selected major lipid loci, i.e., those with lipid associations at $p < 10^{-100}$ based on a score test in the largest European ancestry GWAS. In this context, reproducibility was operationalised as at least one variant from the credible set being associated at $p < 10^{-3}$ based on a score test with serum lipid levels in the target study. We defined the credible set as variants correlated at $r^2 > 0.6$ with the lead SNP from the European discovery study. Correlation was estimated from the 1000 Genomes Project samples with European ancestry. As a benchmark, we also assessed replication in a European ancestry study, UKHLS. We

found evidence of transferability for 76.5% of major HDL loci in this study (Table 1). For the non-European groups rates ranged from 70.6 to 82.4%. Similar reproducibility rates were observed for LDL loci (61.5–76.9%). For major triglycerides (TG) loci, rates ranged from 78.9 to 94.7%, except in APCDR-Uganda. Only 10.5% of the TG loci showed evidence of reproducibility in that sample. Rates for known loci with $p \geq 10^{-100}$ in the discovery set were generally below 10%. However, Biobank Japan, the largest study, exhibited markedly higher reproducibility rates for these loci than the other studies with 24.6–32.7%.

### Trans-ethnic genetic correlations.
Trans-ethnic genetic correlations were estimated between the three largest studies, China Kadoorie Biobank, Biobank Japan and GLGC2013 (Fig. 1). For GLGC2013 and CKB, correlations were 0.999, 0.778, 0.999 for HDL, LDL, and TG, respectively. For GLGC2013 and BBJ, correlations were 0.999, 0.959, 0.961 for HDL, LDL, and TG, respectively. None of the estimates were significantly different from 1 (Supplementary Table 1). We also compared associations across lipid biomarkers. This consistently showed negative genetic correlations between TG associations and HDL associations, with estimates ranging from $r_{gen} = -0.48$ to $r_{gen} = -0.86$.

### Genetic risk scores.
In order to assess patterns of sharing of risk alleles for the smaller studies, we constructed genetic risk scores (GRS) based on the established lipid loci from discovery studies with European-ancestry and assessed the score associations with serum levels of HDL, LDL and TG in HELIC, APCDR-Uganda, CKB and also UKHLS as a benchmark (Fig. 2). All genetic scores were significantly associated with their respective target lipid in the three European samples with largely consistent correlation coefficients and mutually overlapping 95% confidence intervals (CIs) (Table 2). For HDL, LDL and TG, the estimated correlation coefficients ranged from 0.27 to 0.28, 0.23 to 0.28 and 0.20 to 0.24, respectively. In APCDR-Uganda, the strongest association was observed for LDL (r = 0.28, SE = 0.01, $p = 1.9 \times 10^{-107}$ based on a mixed model score test). The HDL association was attenuated compared to the European ancestry samples (r = 0.12, SE = 0.01, $p = 6.1 \times 10^{-22}$). The effect of the TG score was markedly weaker (r = 0.06, SE = 0.01, $p = 4.5 \times 10^{-7}$). For CKB, the HDL GRS had a correlation of $r = 0.18$ (SE = 0.02, $p = 1.4 \times 10^{-22}$) and the LDL GRS of $r = 0.20$ (SE = 0.02, $p = 32 \times 10^{-26}$) while the triglyceride GRS showed a stronger attenuation relative to UKHLS with $r = 0.14$ (SE = 0.02, $p = 3.8 \times 10^{-12}$). We also assessed associations between a given score and levels of each of the other lipid biomarkers (Supplementary Table 2). In line with the trans-ethnic genetic correlation results, we observed inverse associations between the HDL score and TG levels and vice versa in all studies, except APCDR-Uganda.

### Trans-ethnic colocalization.
Differences in LD structure, MAF and sample size make it difficult to assess the transferability of individual loci. Therefore, we propose a new strategy to assess evidence for shared causal variants between two populations: trans-ethnic colocalization. For this we re-purposed a method that was originally developed for colocalization of GWAS and eQTL results: Joint Likelihood Mapping (JLIM)[16]. In order to assess its performance for GWAS results from samples with different ancestry, we carried out a simulation study. UK Biobank (UKB) was used as a reference with European ancestry and compared to CKB and APCDR-Uganda. In order to derive an upper boundary for the power, we compared UKB to the ancestry-matched UKHLS set. Phenotypes were simulated. Effect size estimates were varied between 0.10 and 0.25 in order to represent a range similar to that observed for major lipid loci[15]. In

**Table 1 Percentage of established lipid-associated loci with evidence of reproducibility in target studies**

| P in GLGC: | | $<10^{-100}$ | | | $\geq 10^{-100}$ | | |
|---|---|---|---|---|---|---|---|
| Study | Trait | n.s.[a] | Region[b] | Credible[c] | n.s.[a] | Region[b] | Credible[c] |
| UKHLS | HDL | 5.9 | 17.6 | 76.5 | 81.0 | 13.7 | 5.2 |
| | LDL | 7.7 | 15.4 | 76.9 | 77.0 | 16.4 | 6.6 |
| | TG | 0.0 | 5.3 | 94.7 | 82.1 | 14.5 | 3.4 |
| CKB | HDL | 11.8 | 5.9 | 82.4 | 71.2 | 16.4 | 12.4 |
| | LDL | 7.7 | 30.8 | 61.5 | 83.6 | 7.4 | 9.0 |
| | TG | 5.3 | 15.8 | 78.9 | 82.9 | 10.3 | 6.8 |
| BBJ | HDL | 11.8 | 11.8 | 76.5 | 47.7 | 19.6 | 32.7 |
| | LDL | 7.7 | 30.8 | 61.5 | 64.8 | 10.7 | 24.6 |
| | TG | 5.3 | 10.5 | 84.2 | 55.6 | 12.8 | 31.6 |
| APCDR-Uganda | HDL | 11.8 | 17.6 | 70.6 | 73.2 | 25.5 | 1.3 |
| | LDL | 23.1 | 7.7 | 69.2 | 73.8 | 24.6 | 1.6 |
| | TG | 42.1 | 47.4 | 10.5 | 79.5 | 17.1 | 3.4 |

Only one SNP was kept for each locus with multiple associated variants in close proximity. Regions were defined as 25 Kb either side of the lead variant. The credible set contains the reported lead variant and variants in LD ($r^2 > 0.6$) with it. Results are shown separately for groups of loci by strength of association (whether $p < 10^{-100}$) in the discovery study (GLGC). There were 25, 16, and 27 loci with $p < 10^{-100}$ based on a score test in GLGC for HDL, LDL, and TG, respectively. There were 212, 171, and 158 loci with $p \geq 10^{-100}$ for HDL, LDL, and TG, respectively
[a]No variant in the region associated in target set at $p < 10^{-3}$
[b]No variant in the credible set associated in the target set at $p < 10^{-3}$ but an uncorrelated variant in the region is associated in target set at $p < 10^{-3}$
[c]A variant in the credible set is associated in the target set at $p < 10^{-3}$

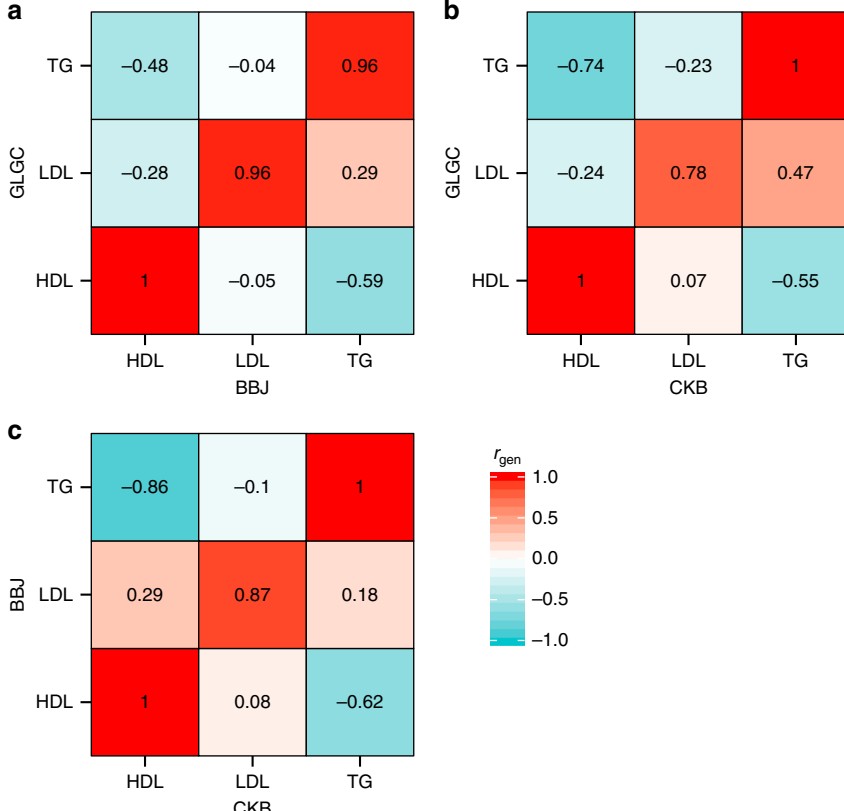

**Fig. 1** Trans-ethnic genetic correlations $r_{gen}$ for associations with high-density lipoprotein (HDL), low-density lipoprotein (LDL) cholesterol and triglycerides (TG). **a** shows the comparison of GLGC2013 (European) and Biobank Japan (BBJ), **b** GLGC2013 and China Kadoorie Biobank (CKB) and **c** BBJ and CKB. The values on the diagonal show correlations for matched lipid markers in the two studies. The off-diagonal values show genetic correlations across lipid biomarkers (e.g. HDL vs TG). Colours correspond to the direction and strength of $r_{gen}$

the simulations of distinct causal variants in the non-European and the reference group, the frequencies of false positives were as expected close to 0.05 (Supplementary Table 3, Supplementary Fig. 1). The power to detect shared associations for betas of 0.25 was 73.1% for APCDR-Uganda, 93.1% for CKB and 0.89 for UKHLS (Fig. 3). To investigate whether the lower power for

APCDR-Uganda could be due to its smaller sample size, we reran the analyses for CKB using a random subset of samples matching the sample size of APCDR-Uganda. For effect sizes <0.2, the results from this analysis revealed decreased detection power relative to the full CKB set but still consistently higher than APCDR-Uganda. This suggests that the power of this trans-ethnic

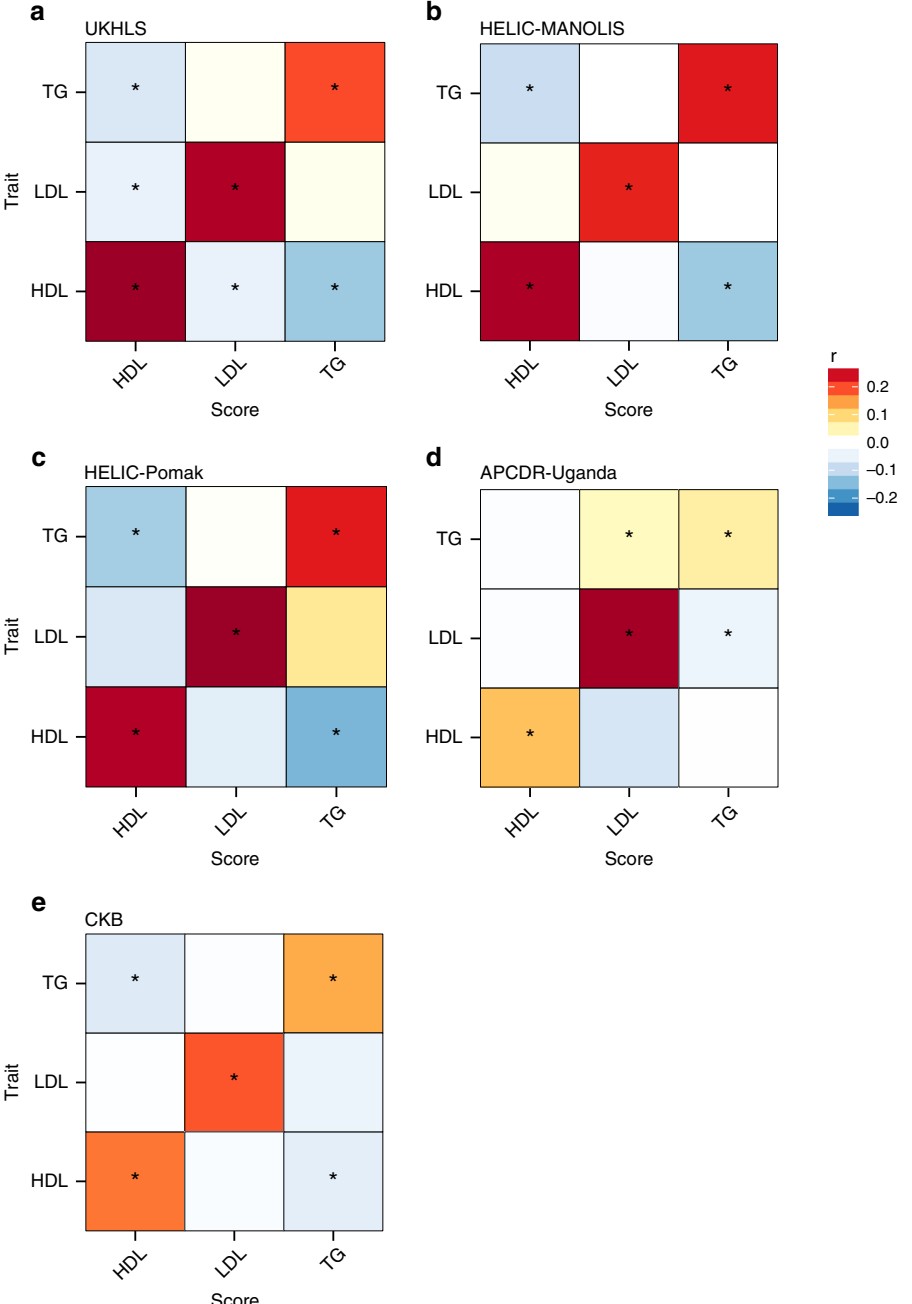

**Fig. 2** Associations of genetic risk scores (GRS) based on established lipid-associated loci with serum levels of high-density lipoprotein (HDL), low-density lipoprotein (LDL) cholesterol and triglycerides (TG) in **a** UKHLS, **b** HELIC-MANOLIS, **c** HELIC-Pomak, **d** APCDR-Uganda, **e** CKB. Estimates are given as correlation coefficients r with colours corresponding to the direction and strength of r. Stars indicate statistically significant associations based on a score test ($p < 0.0056$). The values on the diagonal represent the strength of correlation of a GRS with its target lipid biomarker. The off-diagonal elements show the strength of correlation of a GRS (e.g., TG) with the other lipid markers (e.g., HDL)

colocalization method decreases somewhat with greater genetic distance between the populations that are compared.

We applied trans-ethnic colocalization for established lipid loci to each study with UKHLS as the reference. There was evidence for significant ($p_{jlim} < 0.05$ based on a permutation test) colocalization with at least one of the target studies for about half of the major lipid loci (Supplementary Table 4). For several of the major TG loci, such as 8q24.13, strong evidence of transferability to the Asian studies was observed whilst there was no evidence of association in APCDR-Uganda. Figure 4 shows the regional association plots of this locus for each data set as an

example to demonstrate that differences in LD and frequencies lead to different association patterns. As colocalization can account for such differences, the result from the analysis comparing the European and Asian studies was nevertheless statistically significant ($p < 0.001$).

We compared major lipid loci that showed evidence of transferability to APCDR-Uganda with those that did not. The proximal genes of transferable loci were enriched for lipid pathways including lipoprotein metabolism, lipid digestion mobilisation and transport, chylomicron-mediated lipid transport and metabolism of lipids and lipoproteins. The proximal genes of the non-transferable

**Table 2 Associations of genetic risk scores based on established lipid-associated loci and respective serum lipid levels in UKHLS, HELIC-MANOLIS, -Pomak, APCDR-Uganda, and CKB using a linear mixed model analysis**

| Trait | $N$ | Correlation (SE[a]) | Confidence interval | P-value |
|---|---|---|---|---|
| **UKHLS** | | | | |
| HDL | 9706 | 0.285 (0.010) | 0.265, 0.305 | $4.52 \times 10^{-166}$ |
| LDL | 9767 | 0.274 (0.010) | 0.254, 0.294 | $1.32 \times 10^{-155}$ |
| Triglycerides | 9635 | 0.204 (0.010) | 0.183, 0.223 | $9.62 \times 10^{-87}$ |
| **HELIC-MANOLIS** | | | | |
| HDL | 1186 | 0.279 (0.029) | 0.222, 0.336 | $4.08 \times 10^{-20}$ |
| LDL | 1186 | 0.229 (0.029) | 0.172, 0.286 | $2.41 \times 10^{-14}$ |
| Triglycerides | 1176 | 0.235 (0.030) | 0.176, 0.294 | $4.52 \times 10^{-14}$ |
| **HELIC-Pomak** | | | | |
| HDL | 1078 | 0.268 (0.030) | 0.209, 0.327 | $2.39 \times 10^{-17}$ |
| LDL | 1075 | 0.290 (0.030) | 0.231, 0.349 | $3.04 \times 10^{-19}$ |
| Triglycerides | 1066 | 0.234 (0.030) | 0.175, 0.293 | $2.00 \times 10^{-13}$ |
| **APCDR-Uganda** | | | | |
| HDL | 6407 | 0.121 (0.012) | 0.098, 0.145 | $6.06 \times 10^{-22}$ |
| LDL | 6407 | 0.280 (0.012) | 0.257, 0.304 | $1.91 \times 10^{-107}$ |
| Triglycerides | 6407 | 0.063 (0.013) | 0.038, 0.089 | $4.46 \times 10^{-7}$ |
| **CKB** | | | | |
| HDL | 20810 | 0.180 (0.018) | 0.145, 0.215 | $1.4 \times 10^{-22}$ |
| LDL | 17662 | 0.198 (0.019) | 0.161, 0.235 | $3.2 \times 10^{-26}$ |
| Triglycerides | 20222 | 0.139 (0.020) | 0.100, 0.178 | $3.8 \times 10^{-12}$ |

P-values are based on score tests
[a]SE = standard error

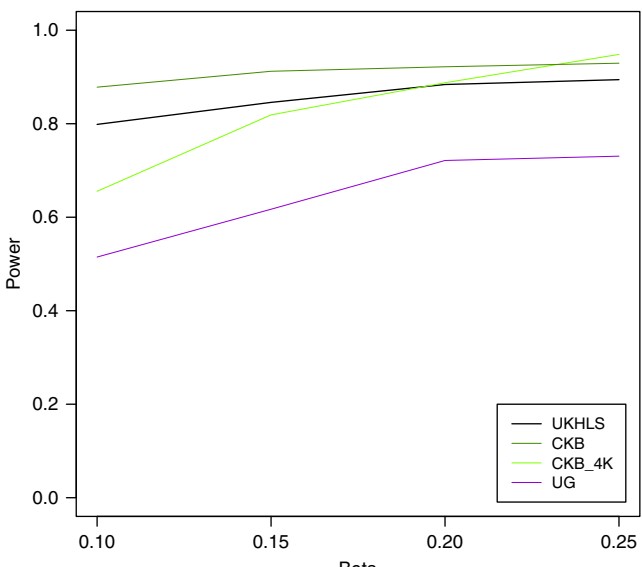

**Fig. 3** Power of trans-ethnic colocalization to detect shared associations for different effect sizes (Beta). Based on a simulation study comparing 50,000 samples with European ancestry from UK Biobank to UKHLS, APCDR-Uganda ("UG"), and China Kadoorie Biobank ("CKB"). Additionally, China Kadoorie Biobank was downsampled to $N = 4597$ ("CKB_4K") to match the sample size of APCDR-Uganda

loci were enriched for several other pathways in addition to lipid metabolism, including SHP2 signalling, ABV3 integrin pathway, cytokine signalling in immune system, cytokine-cytokine receptor interaction and transmembrane transport of small molecules (Supplementary Figs. 2 and 3). We also assessed the associations of these loci with BMI in samples with European ancestry using publicly available summary statistics from the GIANT consortium[17] ($N \geq 484,680$) (Table 3). Ten of the fourteen non-transferable lipid loci had pleiotropic associations with BMI at a Bonferroni-adjusted threshold of $p < 0.0024$. None of the seven transferable lipid loci were associated with BMI.

## Discussion

Recent efforts to increase global diversity in genetics studies have been vital, enabling this comprehensive cross-population comparison of genetic associations with blood lipids. We provide evidence for extensive sharing of genetic variants that affect levels of HDL- and LDL-cholesterol and triglycerides between individuals with European ancestry and samples from China, Japan and Greek population isolates. We estimated that at least about three quarters of major lipid loci are reproducible. This was highly consistent across all studies except for triglyceride loci in APCDR-Uganda. None of the estimates of trans-ethnic genetic correlations between European, Chinese and Japanese samples were significantly different from 1. All GRS associations in the two Greek isolated populations were highly consistent with those in the UK samples (correlations ranged from 0.27 to 0.28, 0.23 to 0.28, and 0.20 to 0.24, for HDL, LDL and TG, respectively, in these studies). Associations of genetic risk scores for LDL were not attenuated in the Ugandan population compared to the UK samples ($r = 0.28$, SE $= 0.01$, $p = 1.9 \times 10^{-107}$ based on a score test).

Previous studies that compared the direction of effect of established loci or assessed associations of genetic risk scores reported differing degrees of consistency[18-29]. However, most of them were conducted in American samples with diverse ancestry, had smaller sample sizes and applied a single-variant look-up or GRS for a limited number of genetic variants. The high degree of consistency for cholesterol biomarkers we observed also contrasts with previously reported trans-ethnic genetic correlations for other traits, such as major depression, rheumatoid arthritis, or type 2 diabetes, which were substantially different from 1[30,31]. In a recent application using data from individuals with European and Asian ancestry from the UK and USA, the average genetic correlation across multiple traits was 0.55 (SE $= 0.14$) for GERA and 0.54 (SE $= 0.18$) for UK Biobank[32].

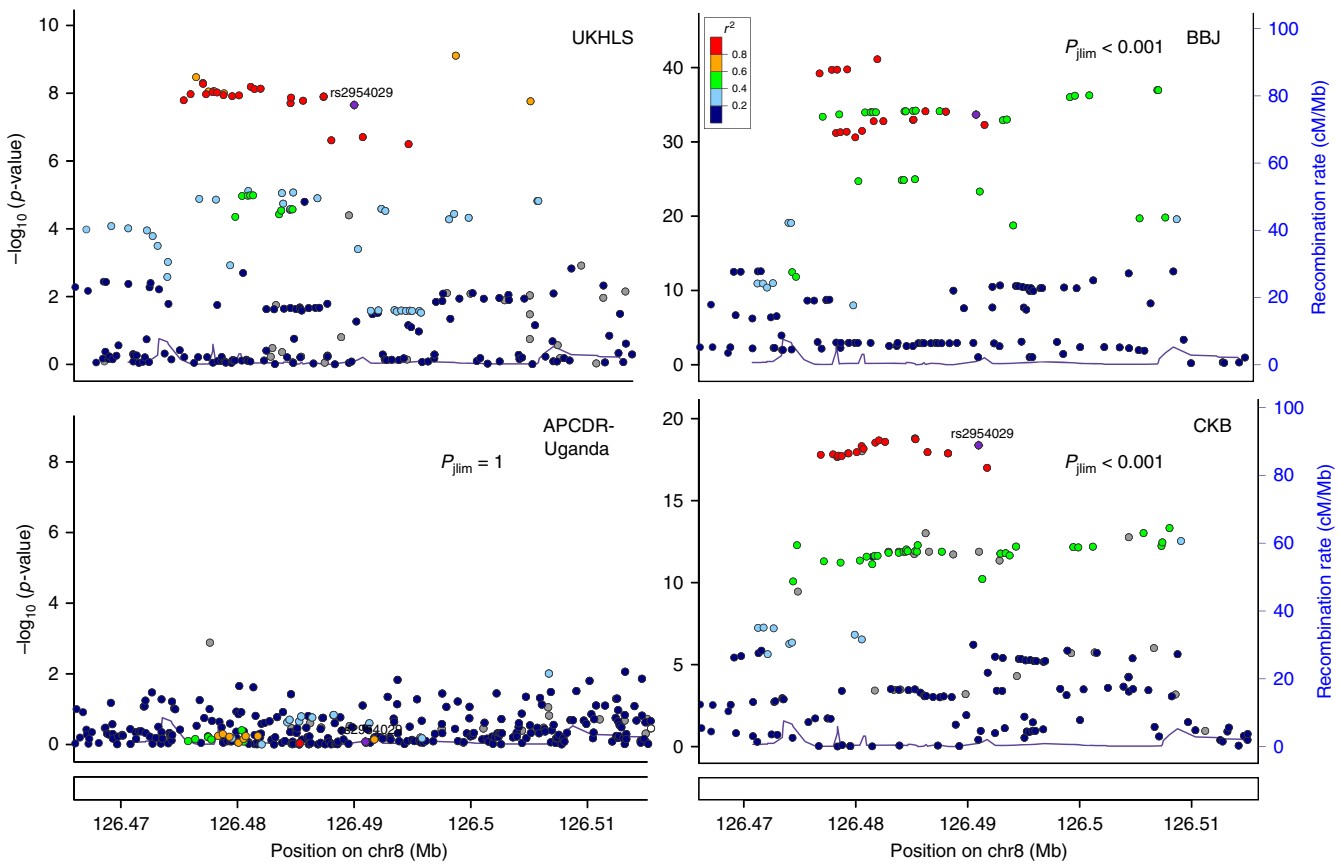

**Fig. 4** Regional plots for SNP associations with triglyceride levels at established triglyceride-associated locus 8q24.13 for UKHLS, Biobank Japan (BBJ), APCDR-Uganda, and China Kadoorie Biobank (CKB) and *p*-value p$_{jlim}$ based on a permutation test for the trans-ethnic colocalization with UKHLS. Filling colour of the points corresponds to the strength of linkage disequilibrium (r$^2$) of each variant with the lead variant rs2954029

**Table 3 Association of established lipid-associated loci with body mass index by whether the locus was transferable to APCDR-Uganda. BMI association results are based on N≥484,680 samples from the meta-analysis between GIANT and UK Biobank[17]**

| Transferable | Trait | rs-id | Chr | Position | Annotation | beta | SE | *P*-value |
|---|---|---|---|---|---|---|---|---|
| No | HDL | rs11755393 | 6 | 34824636 | *UHRF1BP1* | −0.025 | 0.002 | **9.8 × 10⁻⁴⁸** |
| No | HDL | rs1178979 | 7 | 72856430 | *BAZ1B* | −0.010 | 0.002 | **3.1 × 10⁻⁶** |
| No | HDL | rs4731702 | 7 | 130433384 | *KLF14* | 0.008 | 0.002 | **3 × 10⁻⁷** |
| No | HDL | rs2954033 | 8 | 126493746 | *NSMCE2* | −0.010 | 0.002 | **6.4 × 10⁻⁸** |
| No | LDL | rs4245791 | 2 | 44074431 | *ABCG8* | 0.002 | 0.002 | 0.22 |
| No | LDL | rs3846662 | 5 | 74651084 | *HMGCR* | 0.020 | 0.002 | **1.9 × 10⁻³⁵** |
| No | LDL | rs2737229 | 8 | 116648565 | *TRPS1* | 0.014 | 0.002 | **1.9 × 10⁻¹⁵** |
| No | LDL | rs635634 | 9 | 136155000 | *IL6R* | 0.005 | 0.002 | 0.03 |
| No | LDL | rs2000999 | 16 | 72108093 | *HPR* | 0.011 | 0.002 | **8.6 × 10⁻⁸** |
| No | TG | rs1260326 | 2 | 27730940 | *GCKR* | −0.011 | 0.002 | **1.2 × 10⁻¹⁰** |
| No | TG | rs2943641 | 2 | 227093745 | *IRS1* | 0.006 | 0.002 | **5.8 × 10⁻⁴** |
| No | TG | rs6905288 | 6 | 43758873 | *VEGFA* | −0.010 | 0.002 | **1.9 × 10⁻⁹** |
| No | TG | rs11820589 | 11 | 116633862 | *APOA5* | −0.003 | 0.003 | 0.41 |
| No | TG | rs58542926 | 19 | 19379549 | *TM6SF2* | −0.003 | 0.003 | 0.33 |
| Yes | HDL | rs643531 | 9 | 15296034 | *TTC39B* | 0.000 | 0.002 | 0.92 |
| Yes | HDL | rs1800588 | 15 | 58723675 | *LIPC* | −0.002 | 0.002 | 0.25 |
| Yes | HDL | rs3764261 | 16 | 56993324 | *CETP* | −0.002 | 0.002 | 0.39 |
| Yes | HDL | rs16942887 | 16 | 67928042 | *PSKH1* | −0.005 | 0.003 | 0.06 |
| Yes | LDL | rs12740374 | 1 | 109817590 | *CELSR2* | 0.003 | 0.002 | 0.18 |
| Yes | LDL | rs1367117 | 2 | 21263900 | *APOB* | −0.002 | 0.002 | 0.19 |
| Yes | LDL | rs6511720 | 19 | 11202306 | *LDLR* | 0.006 | 0.003 | 0.03 |

Results are shown exclusively for loci where there was clear evidence for or against transferability to APCDR-Uganda (see Methods for more details)
*P*-values in bold indicate SNP associations with BMI that were statistically significant after Bonferroni adjustment

As a limitation of our study, we did not adjust for use of lipid-lowering medication. This could in principle cause a small downward bias for the genetic effect estimates. However, few of the participants of the Ugandan and Chinese studies used lipid-lowering drugs. So this is unlikely to have an effect on the main conclusions of this work.

Differences in LD structure, MAF and sample size make it difficult to assess the transferability of individual loci. We therefore propose a new approach: trans-ethnic colocalization. Simulations showed consistent control of type I error rates, as well as power greater than 80% to detect shared associations between samples with European and Chinese ancestry for SNP effects greater or equal to 0.15. However, power was decreased for comparisons between samples from APCDR-Uganda and UK Biobank (51.5–73.1%). Hence, for the current implementation non-significant colocalization should not be considered as definitive evidence for the absence of shared causal variants when comparing African and European samples. Future work should address this through better modelling of the LD structure. Moreover, for many of the major lipid loci, more than one independent association signal was identified in discovery GWASs[15]. When these are located in close proximity to each other, they can interfere with the trans-ethnic colocalization analysis because JLIM assumes a single causal variant. Therefore, future work should extend this approach to accommodate loci harbouring multiple causal variants.

Using trans-ethnic colocalization, we showed that many established loci for triglycerides did not affect levels of this biomarker in Ugandan samples. This included loci associated at genome-wide significance in all the other studies, such as *GCKR* at 2p23.3 or *LPL* at 8p21.3. The genetic risk score for triglycerides had a weak effect on measured levels in APCDR-Uganda. This is unlikely to be an artefact of unreliable measurement: triglyceride levels had a heritable component in this sample (SNP heritability of 0.25, $SE = 0.05$[8]) and there were genome-wide significant associations. It is also unlikely that this can be explained purely by differences in LD and MAF because they would affect the analyses of the other two lipid biomarkers as well. Instead these discrepancies could be caused by gene-environment interactions. Ten out of fourteen of the lipid loci that were not transferable to the Ugandan samples had pleiotropic associations with BMI in European ancestry samples while none of the transferable loci were linked to BMI. It is possible that the non-transferable variants affect the amount of food intake with downstream consequences for lipid levels. This might require an environment offering diets that are rich in certain nutrients. While the proximal genes for transferable loci were almost exclusively linked to pathways of lipid metabolism, the ones for non-transferable loci were involved in diverse pathways which is in line with this hypothesis. An alternative explanation could be that the non-transferable loci are involved in metabolising nutrients given a particular diet that is not common in Uganda with downstream consequences for weight.

Overall, this could suggest an important role of environmental factors in modifying which genetic variants affect lipid levels. Studying the causes for discordant loci between groups has promise to further elucidate the biological mechanisms of lipid regulation and other complex traits. Applying genetic risk prediction within clinical settings is receiving increasing attention. Our findings demonstrate that the transferability of genetic associations across different ancestry groups and environmental settings should be assessed comprehensively for medically relevant traits. This is important in order to ensure that health benefits of precision medicine are widely shared within and across populations. Ongoing programs in underrepresented countries[33], such as the Human Hereditary and Health in Africa Initiative[34], and programs focussing on underrepresented groups, such as PAGE[35], All of Us[36], or East London Genes and Health[37], could provide the basis for this.

## Methods

**Data resources**. We included data from the Global Lipid Genetics Consortium (European ancestry samples only, GLGC), The UK Household Longitudinal Study (UKHLS), two isolated populations from the Greece Hellenic Isolated Cohorts (HELIC), a rural West Ugandan population from the African Partnership for Chronic Disease Research (APCDR-Uganda) study, China Kadoorie Biobank (CBK), and Biobank Japan (BBJ). Raw genotype and phenotype data were available for UKHLS, APCDR-Uganda, CKB, HELIC-MANOLIS, and HELIC-Pomak. All participants provided written informed consent and each study obtained approval from ethical review boards. The APCDR-Uganda study was approved by the Uganda Virus Research Institute, Science and Ethics Committee (Ref. GC/127/10/10/25), the Uganda National Council for Science and Technology (Ref. HS 870), and the U.K. National Research Ethics Service, Research Ethics Committee (Ref. 11/H0305/5). The HELIC study was approved by the Harokopio University Bioethics Committee. The UKHLS study has been approved by the University of Essex Ethics Committee and the nurse data collection by the National Research Ethics Service (10/H0604/2). For CKB, central ethics approvals were obtained from Oxford University, and the China National CDC. In addition, approvals were also obtained from institutional research boards at the local CDCs in the 10 regions. BBJ was approved by the ethics committees of RIKEN Center for Integrative Medical Sciences and the Institute of Medical Sciences, the University of Tokyo. Our analyses were based on summary statistics for BBJ and GLGC. The details of genotyping, QC and imputation for all studies are summarised in Supplementary Table 5. Descriptive information about the sample sets is provided in Supplementary Table 6. Details of the quality control, imputation, genome-wide association analyses and ethical approval have also been previously described for GLGC[14], BBJ[13], HELIC[10], APCDR-Uganda[8] and UKHLS[12]. Each study confirmed sample ethnicity through PCA which rules out sample overlap between studies.

For CKB, 102,783 participants were genotyped using 2 custom-designed Affymetrix Axiom® arrays including up to 803 K variants, optimised for genome-wide coverage in Chinese populations. Stringent quality control included SNP call rate > 0.98, plate effect $P > 10^{-6}$, batch effect $P > 10^{-6}$, HWE $P > 10^{-6}$ (combined 10df $\chi^2$ test from 10 regions), biallelic, MAF difference from 1KGP EAS < 0.2, sample call rate > 0.95, heterozygosity < mean + 3 SD, no chrXY aneuploidy, genetically-determined sex concordant with database, resulting in genotypes for 532,415 variants present on both array versions. Imputation into the 1,000 Genomes Phase 3 reference (EAS MAF > 0) using SHAPEIT version 3 and IMPUTE version 4 yielded genotypes for 10,276,633 variants with MAF > 0.005 and info > 0.3.

In CKB, lipid levels were regressed against eight principle components, region, age, age², sex, and — for LDL and TG — fasting time² for the single SNP association analysis. For CKB, PCs were included in both single SNP and PRS association analyses to improve inflation. Recruitment for CKB occurred at 10 different rural and urban locations across China leading to somewhat increased population structure. The resulting inflation estimates lambda after PC adjustment were 1.063, 1.050, and 1.053 for HDL, LDL, and TG, respectively. LDL levels were derived using the Friedewald formula. After rank-based inverse normal transformation, the residuals were used as the outcomes in the genetic association analyses using linear regression. Associations were carried out within a mixed model framework using BOLT-LMM[38].

The single SNP association analysis for APCDR-Uganda was carried out within a mixed model framework using GEMMA[39]. Rank-based inverse normal transformation was applied to the lipid biomarkers after adjusting for age and gender. For Uganda, the inflation estimates lambda were 1.000, 1.004, and 1.005 for HDL, LDL, and TG, respectively.

**Established lipid loci**. A list of established lipid-associated loci was extracted from the latest Global Lipid Genetics Consortium (GLGC2017) publication[15] reporting 444 independent variants in 250 loci associated at genome-wide significance with HDL, LDL, and triglyceride levels. We excluded three LDL variants where the association was not primarily driven by the samples with European ancestry. We assessed evidence for transferability of the loci, applied trans-ethnic colocalization and used them to construct genetic risk scores.

**Reproducibility of established lipid loci**. We assessed evidence that these established lipid signals are reproducible in other populations. For loci harbouring multiple signals, we only kept the most strongly associated variant. Out of the 444 loci, this left 170 HDL, 135 LDL and 136 TG variants. We distinguished major loci, i.e. those with $p < 10^{-100}$ based on a score test in GLGC2017. For each lead SNP we identified all variants in LD ($r^2 > 0.6$) based on the European ancestry 1000 Genomes data. We assessed whether the lead or any of the correlated variants, henceforth called credible set, displayed evidence of association in the target study. If this was not the case, we tested whether there was any other variant with evidence of association within a 50 Kb window. We used a p-value threshold of $p < 10^{-3}$ based on a score test. This threshold was derived by computing the minimum p-value in 1000 random windows of 50 Kb for each study. Less than 5% of random windows had a minimum $p < 10^{-3}$ for the

non-European ancestry studies. While this p-value threshold might not be appropriate to provide conclusive evidence of reproducibility for individual loci, we used this to test evidence of reproducibility across sets of loci. These analyses excluded the HELIC studies because the smaller sample size makes it difficult to differentiate between lack of power and lack of reproducibility.

**Trans-ethnic genetic correlations**. We used the popcorn software[30] to estimate trans-ethnic genetic correlations between studies while accounting for differences in LD structure. This provides an indication of the correlation of causal-variant effect sizes across the genome at SNPs common in both populations. Variant LD scores were estimated for ancestry-matched 1000 Genomes v3 data for each study combination. The estimation of LD scores failed for chromosome 6 for some groups. We therefore left out the major histocompatibility complex (MHC) region (positions 28,477,797 to 33,448,354) from chromosome 6 from all comparisons. Variants with imputation accuracy $r^2 < 0.8$ or MAF $< 0.01$ were excluded. Popcorn did not converge for any of the studies with less than 20,000 samples. Therefore, results are presented for comparisons between GLGC2013, CKB and BBJ. We estimated effect rather than impact correlations. We used a Bonferroni correction to adjust for multiple testing of three traits with each other ($p < 0.05/9 = 0.0056$).

**Genetic risk scores**. As it was not possible to compute trans-ethnic genetic correlations for UKHLS, the HELIC cohorts, and APCDR-Uganda, we created genetic risk scores based on the established lipid loci and assessed their associations with serum lipid levels in these studies. We also tested the associations of GRS in CKB as raw data were available for this study as well. Age and sex were adjusted for by regressing them on the lipid biomarker values and using the residuals as outcomes for subsequent analyses. For CKB, we additionally adjusted for 20 PCs and region covariates in order to ensure population structure was accounted for. To ensure values are normally distributed, we used rank-based inverse normal transformation for all biomarkers and data sets which involves ordering values first and then assigning them to expected normal values. To make sure GRS were comparable across studies, we excluded variants that were absent, rare (MAF $< 0.01$) or badly imputed ($r^2 < 0.8$) in any of the studies and variants that had different alleles from those in the GLGC. The variant with larger discovery p-value from each correlated pair of SNPs ($r^2 > 0.1$) was also removed. These filters were applied based on each, UKHLS, HELIC, and APCDR-Uganda and then the intersection of variants was carried forward to generate GRS. Out of the 444 loci, this left 120, 103, and 101 variants for HDL, LDL and TG, respectively (Supplementary Table 7). We created trait-specific weighted GRS. The β-regression coefficients from SNP-trait associations in GLGC2017[15] were used as weights. All lipid biomarkers and scores were scaled to mean $= 0$ and standard deviation $= 1$ for each study, so that the regression coefficients represent estimates of the correlation between scores and lipid biomarkers.

We carried out association analyses between each genetic risk score and each lipid biomarkers using a linear mixed model with random polygenic effect implemented in GEMMA[39] in order to account for relatedness and population structure. For CKB, we used BOLT-LMM because it is efficient for large samples. We used a Bonferroni correction to adjust for multiple testing of three GRS with three different lipid biomarker outcomes ($p < 0.05/9 = 0.0056$ for the score test).

**Trans-ethnic colocalization**. Differences in allele frequency, LD structure and sample size make it difficult to assess whether a given GWAS hit is transferable to samples with different ancestries. Therefore, we applied trans-ethnic colocalization. Colocalization methods test whether the associations in two studies can be explained by the same underlying signal even if the specific causal variant is unknown. The joint likelihood mapping (JLIM) statistic was developed by Chun and colleagues to estimate the posterior probabilities for colocalization between GWAS and eQTL signals and compare them to probabilities of distinct causal variants[16]:

$$\Lambda = \sum_{i \in N_\theta^1(m^*)} L_1(i) \times \log \frac{L_1(i)L_2(i)}{\max_{j \notin N_\theta^2(i)} L_1(i)L_2(j)} \quad (1)$$

Where $i$ SNP1; $m^*$ lead SNP; $L_1(i)$ likelihood of SNP i being causal for trait 1; $L_2(i)$ likelihood of SNP i being causal for trait 2; $N_\theta^1(i)$, $N_\theta^2(i)$ sets of SNPs in LD with $i$; $\theta$ LD threshold.

JLIM explicitly accounts for LD structure. Therefore, we assessed whether it is suitable for trans-ethnic colocalization. For the reference sample set, it was possible to use genome-wide summary statistics for the analysis. For this set, LD scores were estimated using a subset of samples from the 1000 Genomes Project v3 that had matching ancestry to that study. The second sample set needed raw genotype data and LD was estimated directly for these samples. JLIM assumes only one causal variant within a region in each study. We therefore used small windows of 50Kb for each known locus to minimise the risk of interference from additional association signals. Distinct causal variants were defined by separation in LD space by $r^2 \geq 0.8$ from each other. We excluded loci within the MHC region due to its complex LD structure. We used a significance threshold of $p < 0.05$ given the evidence of association of the established lipid loci in Europeans and the overall evidence for shared causal genetic architecture across populations for most lipid traits from our other analyses. We compared each target study to UKHLS because of the study's matched ancestry with

the discovery study, high level of homogeneity in terms of ancestry, biomarker quantification and study design.

**Simulation**. To test the power of trans-ethnic colocalization to detect associations shared between pairs of populations with different ancestry, we ran JLIM on two sets of simulated traits with realistic effect size and environmental noise level. The first set of simulations used the same causal variant in both populations, whereas the second set of simulations used discordant causal variants. Causal variants were selected using the sample function in R, corresponding to a uniform random draw from the entire chromosome. We sampled 10,000 randomly chosen biallelic variants with MAF $> 0.05$ and simulated random phenotypes in UKHLS, CKB, APCDR-Uganda and 50,000 individuals with British ancestry from UK Biobank as the reference set. For UK Biobank we applied the QC and used the ancestry assignment provided by Bycroft et al.[40]. UKHLS was included as an ancestry-matched set in order to derive an upper limit estimate of the power. For each data set relatives were excluded. We also sub-sampled CKB to match the number of individuals in APCDR-Uganda in order to test whether the difference in performance was due to ancestry or sample size. We used a simple linear model to generate the phenotype for each individual $i$:

$$y_i = \beta * (x_i - 1) + \eta_i \quad (2)$$

where y is the phenotype value, $\beta$ is the effect size, $x$ is the number of the alternate alleles carried at the locus and $\eta_i \sim N(0, \sigma^2)$, where $\sigma^2$ is the variance of the environmental noise and $Cov(\eta_{1i}, \eta_{1j}) = 0$. We tested effect size estimate beta from 0.10, 0.15, 0.20, and 0.25 in order to represent a range similar to that observed for the major lipid loci[15]. We used $\sigma^2 = 1$ to match the trait variances of the standardised phenotypes.

**Comparison of transferable loci with non-transferable loci**. We assessed whether there are any systematic differences between loci that are shared between European ancestry samples and APCDR-Uganda and loci that are not. We identified all loci with evidence of reproducibility based on the above definition that also had significant ($p < 0.05$) colocalization based on a permutation test. We only kept one variant per region. We contrasted them with loci where none of the evidence suggested generalisation: $p > 0.05$ for colocalization or missing result due to failed convergence, no variant with a lipid association at $p < 10^{-3}$ in the region and the lead variant from the discovery study was not rare in APCDR-Uganda. We identified the nearest protein coding gene for each locus and carried out pathway analyses for the two sets using FUMA[41]. We also assessed the associations of the lead variants with body mass index (BMI) in European ancestry samples using results from a meta-analysis between the GIANT consortium and UK Biobank[17]. We used a Bonferroni adjusted p-value threshold.

**Reporting summary**. Further information on research design is available in the Nature Research Reporting Summary linked to this article.

## Data availability

The UKHLS EGA accession number is EGAD00010000918. Genotype-phenotype data access for UKHLS is available by application to Metadac (www.metadac.ac.uk). Summary statistics for GLGC (http://csg.sph.umich.edu/abecasis/public/) and Biobank Japan (http://jenger.riken.jp/en/) are publicly available. The HELIC genotype and WGS datasets have been deposited to the European Genome-phenome Archive (https://www.ebi.ac.uk/ega/home): EGAD00010000518; EGAD00010000522; EGAD00010000610; EGAD00001001636, EGAD00001001637. The APCDR committees are responsible for curation, storage, and sharing of the APCDR-Uganda data under managed access. The array and sequence data have been deposited at the European Genome-phenome Archive (EGA, http://www.ebi.ac.uk/ega/, study accession number EGAS00001000545, datasets EGAS00001001558 and EGAD00001001639 respectively) and can be requested through datasharing@sanger.ac.uk. Requests for access to phenotype data and summary statistics may be directed to data@apcdr.org. This is restricted to research-related purposes. Uploading and sharing of individual genetic data from CKB are subject to restrictions according to the Interim Measures for the Administration of Human Genetic Resources administered by the Human Genetic Resources Administration of China (HGRAC). Summary data including allele frequencies and GWAS summary statistics are available on application. This is restricted to research-related purposes. Other individual-level CKB data are available through www.ckbiobank.org, subject to completion of a Material Transfer Agreement, either through Open Access or on application. CKB data access is subject to oversight by an independent Data Access Committee.

## Code availability

Our code to run trans-ethnic colocalization using JLIM and simulations is available through github: https://github.com/KarolineKuchenbaecker/TEColoc

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

## Acknowledgements

C.K.B. thanks the participants, project staff, the China National Centre for Disease Control and Prevention and its regional offices. The Chinese National Health Insurance scheme provided electronic linkage to all hospital admission data. We thank the residents of the Pomak villages and of the Mylopotamos villages for taking part. We thank the African Partnership for Chronic Disease Research (APCDR) for providing a network to support this study as well as a repository for deposition of curated data. We also thank all study participants who contributed to this study. UKHLS is led by the Institute for Social and Economic Research at the University of Essex and funded by the Economic and Social Research Council. The survey was conducted by NatCen and the genome-wide scan data were analysed and deposited by the Wellcome Trust Sanger Institute. This work was funded by the Wellcome Trust (WT098051), (212360/Z/18/Z), and the European Research Council (ERC-2011-StG 280559-SEPI). The baseline survey and first resurvey for CKB were supported by a research grant from the Hong Kong Kadoorie Charitable Foundation. Longterm followup and the second resurvey were supported by grants from the UK Wellcome Trust (212946/Z/18/Z, 202922/Z/16/Z, 104085/Z/14/Z, 088158/Z/09/Z), National Natural Science Foundation of China (81390540, 81390541, 81390544), and National Key Research and Development Program of China (2016YFC 0900500, 0900501, 0900504, 1303904). DNA extraction and genotyping was supported by grants from GlaxoSmithKline and the UK Medical Research Council (MCPC13049, MCPC14135). M.V.H. is supported by the British Heart Foundation (FS/18/23/33512) and the National Institute for Health Research Oxford Biomedical Research Centre. The British Heart Foundation, UK Medical Research Council, and Cancer Research UK provide core funding to the Clinical Trial Service Unit and Epidemiological Studies Unit, Oxford University (Oxford, UK). APCDR-Uganda was funded by the Wellcome Trust, The Wellcome Trust Sanger Institute (WT098051), the UK Medical Research Council (G0901213-92157, G0801566, and MR/K013491/1), and the Medical Research Council/ Uganda Virus Research Institute Uganda Research Unit on AIDS core funding. The UK Household Longitudinal Study was funded by grants from the Economic & Social Research Council (ES/H029745/1) and the Wellcome Trust (WT098051).

## Author contributions

K.K. conceived this project and supervised the work. K.K. and N.T. carried out the genetic correlation and PRS analyses. K.K. carried out all analyses involving trans-ethnic colocalization. K.K. wrote the manuscript. K.K. and A.E. carried out the simulation study. T.R. implemented earlier versions of the genetic risk scores. HELIC: E.Z. and G.D. are the principle investigators, A.G. and L.S. carried out the quality control, M.K. and E.T. were involved in data collection. APCDR-Uganda: M.S., D.G., G.A., J.S., A.K. were involved in collecting and preparing data as well as leading the study. China Kadoorie Biobank: Z.C. and L.L. are the principle investigators; R.G.W. is the genomics lead; R.G.W., I.Y.M., H.D., Y.G. and M.V.H. were involved in data collection; R.G.W. and K.L. carried out

quality control and genome-wide association analysis for lipid biomarkers; K.L. carried out the genotype imputation. All authors approved the manuscript.

## Additional information

**Competing interests:** The authors declare no competing interests.

## Understanding Society Scientific Group

Michaela Benzeval[25], Jonathan Burton[25], Nicholas Buck[25], Annette Jäckle[25], Heather Laurie[25], Peter Lynn[25], Stephen Pudney[25], Birgitta Rabe[25] & Dieter Wolke[26]

[25]Institute for Social and Economic Research, University of Essex, Wivenhoe Park, Colchester CO4 3SQ, UK. [26]Department of Psychology, University of Warwick, Coventry CV4 7AL, UK

