## [Peer Review File · Nature Communications]

Reviewers' comments:

Reviewer #1 (Remarks to the Author):

The authors aim to quantify the generalizability of established lipid loci previously-identified in primarily European-descent populations. To accomplish this aim, they used existing genetic resources in one sub-Saharan African, two East Asian, and two Greek isolate populations along with publicly available summary statistics from previous studies by the GLGC consortium. They assessed genetic correlation of associations across traits and studies for single variants and PRS. The authors also applied established trans-ethnic colocalization methods in a novel fashion to assess generalization. The authors found that a large majority of the loci examined generalized across populations, with a few exceptions for the African population. The authors aims are timely as there is an immediate need to improve risk prediction across global populations in order to advance precision medicine initiatives. Additionally, I appreciate the innovative application of the established transethnic colocalization method. However, despite these positives, I have many concerns about the study design, results, and interpretation of that significantly decrease the potential impact of the study findings.

Major Concerns

-It is unclear in the results and methods what SNPs were used to construct the PRS. What reference was used for the LD pruning when selecting variants? Was the same LD reference used for all studies? Did the investigators start with the 444 tag variants from the GLGC, or did they start with the credible set of SNPs from each locus?

-It is not clear why only a subset of the study populations were used for the PRS and transethnic genetic correlations analyses. The authors state that for small populations with $N < 20,000$, popcorn analyses did not converge. However, an N of 20,000 is not a small sample, thus the method may not be appropriate here. Did the authors evaluate alternative strategies to better utilize their datasets? Also, this does not explain why the studies with $N > 20,000$ were not included in the PRS analyses.

-The description of the trans-ethnic colocalization results is confusing and lacks important details. The authors refer only to Figure 3b and c, but not 3a and d. What is the purpose of Figure 3b and c? Also, panel d in Figure 3 does not follow the same pattern as the other figures, but this is not explained in the figure heading or in the main text (i.e. sub-panels for other sections contain locuszoom plots for UK, Japan, Uganda, and China; however, d has only Japan and Uganda and highlights two SNPs in the same region).

-The authors do not mention the use of eMERGE data in the Introduction, Results, and Discussion sections; yet, eMERGE is present in select tables and mentioned in the methods. Why is data from this consortium used and not interpreted?

-There is no mention of correcting lipid values for medication use. Was this data available?

-The authors excluded all of chromosome 6 from genetic correlation analyses stating that LD calculations failed. How many known loci were excluded as a result? Did LD calculations fail for all ancestry-matched references, or just some? Did the authors try to exclude highly variable regions of chr 6 before excluding the whole chromosome?

-There is no description of genotyping or imputation (arrays used, QC procedures, exclusions, imputation methods, references used).

Minor Concerns

-When reporting the results, the authors are often vague (e.g. "correlations ... were close to 1", "largely consistent correlation", "correlations were high"). It would help to report specific values and ranges within the text to assist the reader in following the interpretation and discussion of results.

-The authors use the term causal, but they are not utilizing methods that determine causality. Instead

of referring to causal variants, refer to associated loci, or tag/lead SNPs underlying loci.

-Minor typos throughout the article (e.g. in Introduction, fifth line, "representation" should be "represent"; "und" on page 14, etc.)

-The authors use the term replication, but generalization is more appropriate for the non-European descent populations.

-The authors mention overlapping confidence intervals in results, but only report SE in tables and do not supply any figures that illustrate this. While one can calculate the CI from the data provided, it would make it easier if these numbers were provided for easy assessment (i.e. in forest plots).

-Section headers in the results would be beneficial.

-The authors state that they obtained publicly available BMI data from Pulit et al.; however, the work referenced was on WHRdjBMI. I believe they meant to cite Yengo et al. (PMID: 30124842).

-Manhattan plots are not necessary.

-The authors adjust for principal components in the SNP associations, but do not mention them in the PRS analyses.

-A brief description of what defines a credible set would be helpful in the Results in addition to the methods to assist in interpretation.

-The authors do not justify the use of different transformation methods used across studies. Why not harmonize the phenotypes.

-Why did the authors choose not to meta-analyze the HELIC studies? Is there prior evidence of population stratification?

-The equation for trans-ethnic colocalization is missing or did not transfer during pdf creation.

-The figure headings are not descriptive enough (i.e. off-diagonal elements are not defined in Figure 1, Table 4 should explain why so few loci are represented, Figure 2 is not very informative, etc.).

Reviewer #2 (Remarks to the Author):

This study investigated sharing genetic variants of major lipids profile across different ethnic groups by several strategies. First, they estimated replication rates of lipid biomarkers by selecting major lipid loci (with $p < 10^{-100}$) as credible set and defined replication with $p < 0.001$ in target cohorts. Secondly, they estimated genetic correlations between cohorts using trans-ethnic genetic correlation method for large studies and polygenic score association for smaller cohorts. Then they applied trans-ethnic colocalization to identify shared lipids loci by different ethnic groups. The aims of the study are clear and the paper is well organized. However, the following points should be addressed.

Major comments:

The interesting results presented may indicate that the genetic components of TG in African Americans might be distinct but major genetic variants of LDL and HDL share in some degree across different ethnic groups. However, the results were not discussed for the possible variation of lipids profile in this population. The lipids values are on average lower comparing with other populations (Supplementary Table 4). What is the phenotypic correlation between TG and HDL? The correlation is probably low according to the results in Supplementary Table 2. If it is true, any mechanism might affect shared genetic components of TG should be investigated or discussed.

It is confusing that the authors used different datasets/cohorts for different strategies. The purpose of the paper is to quantify genetic sharing among populations from Europe/North America, Asia and Africa. As the authors mentioned, they have both phenotype and genotype data available from eMERGE, UKHLS, CKB, HELIC, and APCDR-Uganda and summary statistics from GLGC and BBJ. The

authors tested replication rates with all cohorts except for HELIC. For genetic correlation estimation and trans-ethnic colocalization, the authors excluded eMERGE. Explanations are needed for excluding specific cohorts from different analysis purposes.

According to the sample size limitation, the authors can only estimate trans-ethnic genetic correlations for some large cohorts. The polygenic scores were alternatively used to test sharing genetic correlations among smaller ethnic groups. In order to have better understanding and comparison for genetic correlations, it will be helpful to include all available cohorts for the associations with polygenic scores.

The novel part of this study was to use the trans-ethnic colocalization to identify shared loci accounting for LD structure in different ethnic groups. However, they need a clear description and consistency of the results. For example, the p values in Figure 3 are not consistent with those in Supplementary Table 3. The Result in Page 6, Paragraph 3, Line 4 indicates LPL loci but shows ABC1A loci in Figure 3c. More explanations of results from trans-ethnic colocalization, ie, Figure 3a and 3d, will be necessary. It seems that the regional plots of Japan and China show a similar pattern but the p values are significant different.

The definition of polygenic scores and genetic risk scores is sometimes confusing. According to the paper by Khera et al (Nature Genetics 2018), it is more suitable to refer polygenic scores using whole genome data and genetic risk scores by GWAS significant SNPs.

The authors used eMERGE data but didn't mention it in the Introduction.

Because of the multiple testing issue, what is the evidence for the significant threshold definition, ie, $p < 0.001$ for replication, and $p < 0.05$ for trans-ethnic colocalization?

Minor comments:

Table 1: please add the numbers of SNPs with $p < 10^{-10}$ and $p > 10^{-10}$, respectively.

Please use consistent names for all cohorts. For example, I assume 'UG' in Table 1 should be 'APCDR-Uganda'. The Figure 3 shows each cohort with the country name.

In Page 10, Paragraph 1, Line 6, the study details in 'Supplementary Table 3' should refer to 'Supplementary Table 4'.

Reviewer #3 (Remarks to the Author):

Telkar et al. present an interesting and timely study on the degree with which the genetic architecture of lipid levels is shared between populations of different ancestry. The paper is well written, and relies on impressive amounts of datasets and statistical analyses. IT useS state-of-the-art software and propose a modified version of JLIM for identifying shared genetic effects across populations with different ancestries. However, the datasets and analyses used in the paper are in my opinion not always described with sufficient detail. I understand that this is difficult because of the large number of different analyses and datasets used in the paper, but they are necessary for me to review the study and for anyone wanting to replicate it. Indeed, the devil is in the details. Also, although the extension of JLIM to detecting shared genetic effects between ancestries is very interesting, and I would highly encourage the authors to pursue these ideas further, I am not convinced it is working as intended based on the simulations. However, the JLIM results are in my opinion just a minor point in

this study, and I think it is still interesting as it indicates that the genetic architecture of lipid levels is somewhat consistent between populations.

Comments:

- There's no description of how you processed the APCDR-Uganda, eMERGE, and CKB data, which I presume you performed some QC on. E.g., did you filter out individuals of non-Asian ancestry in the Asian cohorts? Similarly, did you filter out individuals of with non-European ancestry in the European eMERGE, etc. A PCA plot for these cohorts, e.g. a projection to the PCA plot from 1000 genomes samples would clearly reveal any admixed individuals. Did you filter out SNPs, or perform relatedness filtering on the samples? All of this information is important to understand how to interpret the results. It would allow you to rule out any possible association signals due to admixed individuals. If this information is left out here because the data was already processed and QC'd in another study, I would appreciate if the specific citations in question were clearer. Similarly, I could not find information on how you QC or select British individuals from the UKB. I am sure this is all trivial or standard, but I would appreciate some descriptions. Finally, regarding the HELIC and UKHLS datasets, (although perhaps not necessary) I would appreciate if you verified the ancestry filtering in these, e.g. by projecting them onto a 1000 genomes PCA plot.
- Is there any chance of sample overlap between these studies and the GLGC2017 study? Please clarify. If there is overlap, then the correlations will be (partially) driven by overfitting.
- Regarding JLIM simulations, how about using the same simulation setup to test for trans-ethnic colocalizations in a European dataset. This would represent the best possible case, a scenario, representing an upper limit on your statistical power. Indeed, given the huge simulated genetic effect (phenotypic variance explained is likely between 1-20%, depending on MAF), the statistical power does not depend on the sample size differences between samples (as it's probably near maximum) but rather on LD differences. In which case, I am worried that the model may not be able to detect shared associations even if they are true (due to LD), leading to downwards biased colocalization frequency estimates. Perhaps you could address this in the discussion, or better yet, extend simulations to study this further (although that's possibly better suited for future studies where you also study the effect of multiple loci, etc.).
- p. 13, The modified version of JLIM uses LD scores that are estimated using ancestry matched samples from the 1000 Genomes Project v3. However, it is unclear to me how you actually do this. Did you match the ratio of ancestry in your data, say 50% EUR, 20% Uganda and 30% Chinese with similar ratios in the 1000 genomes data, and use that to estimate LD? Which 1000 genomes population did you use as representatives? I would prefer more details here.

Minor comments:

- On p. 5, "biomarkers" probably refers to lipids. Is that true? This is possibly confusing.
- On p. 5, you "estimated" the PRS associations in HELIC, APCDR-Uganda and also UKHLS. Instead of "estimated" would prefer validated, or predicted to somehow make clear the distinction between training and validation (which I assume this is).
- Regarding supplementary figure 1, how about using a QQ-plot instead, and possibly summarizing in a mean/median p-value. You can also test for significance using a Kolmogorov-Smirnov test.
- On p. 7, first paragraph in Discussion. I suggest to drop the parenthesis around "triglyceride loci except in APCDR-Uganda", because this is an important result.
- On p. 10. Why does eMERGE suddenly show up in the Methods section, when it's not even mentioned in the results or introduction?
- p. 10, first paragraph. You probably mean supplementary table 4, not 3.
- Why use GEMMA for GWAS in APCDR-Uganda and BOLT-LMM for eMERGE? I would prefer greater consistency in your choices of methods and analyses.
- In the GWAS, did you add some covariates to the analyses, e.g. PCs, sex, age? Please avoid heritable covariates (see Aschard et al., AJHG 2015).

- I would appreciate if you clarified what you mean by “rank-based inverse normal transformation”, e.g. using an equation or two. I assume it means you ranked the lipid levels, and assumed the original values followed a Gaussian distribution, and then obtained the corresponding z-scores. Also, why do you later use a Box-Cox transformation? Again, I would recommend greater consistency in choices of analyses methods.
- On p. 12. Why was it not possible to compute transethnic genetic correlations for the UKHLS, the HELIC cohorts, and APCDR-Uganda data? I would appreciate more details on this.
- When calculating PRS, I wonder how the r^2 threshold affects predictions when predicting into individuals of different ancestry than the training data had. Perhaps predictions improve if you increased the threshold to 0.2 (or larger values)?
- I liked the fact that you used mixed linear models to account for relatedness in the PRS correlations. Would it make sense to add a PCs as covariates to the model as well, for robustness (in case there are environmental factors that correlate with PCs)? I'm also curious, whether one would see any association with PCs (or 1000 genomes PCs projections) in these populations?
- p. 13, second paragraph, you mention that you used summary statistics, possibly implying that you also run it on samples with individual-level genotypes? I think the formulation could be improved.
- p. 13, last paragraph. You chose the causal variant “randomly”, but how exactly? Uniformly from the same region/gene, or the entire genome?
- Table 1., you refer to the APCDR-Uganda data as UG, but do not point this out in the caption, or use UG anywhere else in the paper.

Bjarni J Vilhjalmsón

Response to referees for NCOMMS-19-09413-T

Reviewers' comments

Reviewer #1 (Remarks to the Author):

The authors aim to quantify the generalizability of established lipid loci previously-identified in primarily European-descent populations. To accomplish this aim, they used existing genetic resources in one sub-Saharan African, two East Asian, and two Greek isolate populations along with publicly available summary statistics from previous studies by the GLGC consortium. They assessed genetic correlation of associations across traits and studies for single variants and PRS. The authors also applied established trans-ethnic colocalization methods in a novel fashion to assess generalization. The authors found that a large majority of the loci examined generalized across populations, with a few exceptions for the African population. The authors aims are timely as there is an immediate need to improve risk prediction across global populations in order to advance precision medicine initiatives. Additionally, I appreciate the innovative application of the established transethnic colocalization method. However, despite these positives, I have many concerns about the study design, results, and interpretation of that significantly decrease the potential impact of the study findings.

We would like to thank the reviewer for the accurate summary of our work and for highlighting its importance.

Major Concerns

1) -It is unclear in the results and methods what SNPs were used to construct the PRS. What reference was used for the LD pruning when selecting variants? Was the same LD reference used for all studies? Did the investigators start with the 444 tag variants from the GLGC, or did they start with the credible set of SNPs from each locus?

The starting point for constructing the GRS (PRS renamed GRS, see Reviewer 2, comment 5) were indeed the 444 lead variants identified by GLGC2017. In order to make the GRS associations more comparable across populations, we created a subset of variants that are sufficiently frequent, well imputed and consists only of independent SNPs in each of the target data sets. Therefore, we applied LD filtering using each of the raw data sets rather than a reference panel which would indeed have been problematic because it might not be representative for all of the groups. We have updated the methods section to clarify this (p 12):

"To make sure GRS were comparable across studies, we excluded variants that were absent, rare ($MAF < 0.01$) or badly imputed ($r^2 < 0.8$) in any of the studies and variants that had different alleles from those in the GLGC. The variant with larger discovery p -value from each correlated pair of SNPs ($r^2 > 0.1$) was also removed. These filters were applied based on each,

UKHLS, HELIC, and APCDR-Uganda and then the intersection of variants was carried forward to generate GRS."

2) -It is not clear why only a subset of the study populations were used for the PRS and transethnic genetic correlations analyses. The authors state that for small populations with $N < 20,000$, popcorn analyses did not converge. However, an N of 20,000 is not a small sample, thus the method may not be appropriate here. Did the authors evaluate alternative strategies to better utilize their datasets? Also, this does not explain why the studies with $N > 20,000$ were not included in the PRS analyses.

Most methods to compute parameters of genetic architecture require very large sample sizes and sufficient precision for effect estimates across the genome. Therefore, it was not possible to achieve convergence for HELIC, APCDR-Uganda or UKHLS with the Popcorn algorithm to estimate trans-ethnic genetic correlations. We have recently carried out a review of methods for ancestrally diverse samples and found that no other algorithm for genetic correlations exists that could be used in this context. Therefore, we applied GRS as a feasible alternative in this context.

It is true, however, that the GRS approach does not need to be restricted to the smaller studies. We have now tested the GRS associations in China Kadoorie Biobank and report the methods in page 13 and the results in updated Table 2 and on page 6:

"For CKB, the HDL GRS had a correlation of $r=0.18$ ($SE=0.02$, $p=1.4 \times 10^{-22}$) and the LDL GRS of $r=0.20$ ($SE=0.02$, $p=32 \times 10^{-26}$) while the triglyceride GRS showed a stronger attenuation relative to UKHLS with $r^2=0.14$ ($SE=0.02$, $p=3.8 \times 10^{-12}$)."

Calculating genetic risk scores and assessing their associations requires access to raw genotype and phenotype data. Therefore, we were not able to apply this method to Biobank Japan.

3) -The description of the trans-ethnic colocalization results is confusing and lacks important details. The authors refer only to Figure 3b and c, but not 3a and d. What is the purpose of Figure 3b and c? Also, panel d in Figure 3 does not follow the same pattern as the other figures, but this is not explained in the figure heading or in the main text (i.e. sub-panels for other sections contain locuszoom plots for UK, Japan, Uganda, and China; however, d has only Japan and Uganda and highlights two SNPs in the same region).

Figure 3 (now figure 4) intended to make several points. While a) to c) were examples of different patterns of transferability, d) demonstrated that the method can fail when more than one causal variant is present at a locus. We appreciate that this is too complicated and have now simplified the figure by presenting only one example for the correspondence between colocalization results and visible association patterns for each of the four groups (Figure 4 on p31).

4) -The authors do not mention the use of eMERGE data in the Introduction, Results, and

Discussion sections; yet, eMERGE is present in select tables and mentioned in the methods. Why is data from this consortium used and not interpreted?

We originally included eMERGE as a second independent data set with European ancestry. As described in the methods section, we used it to test whether the reproducibility rates for established loci in a matched ancestry sample are consistent across different European ancestry samples. The rates were highly consistent between UKHLS and eMERGE as well as most of the data sets for other ancestry groups. Therefore, we have now removed eMERGE from this manuscript because the respective results do not significantly contribute to the findings.

5) -There is no mention of correcting lipid values for medication use. Was this data available?

Our analyses did not adjust for medication. For the rural Ugandan population, lipid lowering medication is unlikely to be widely used. Similarly, lipid-lowering medication was not in common use amongst the study population of China Kadoorie Biobank. At baseline, only 4.7% of those with a history of cardiovascular disease (0.2% of the cohort) were currently taking statins. The summary statistics for the European reference data were based on a mix of studies some of which excluded participants on lipid lowering medication and some that did not¹. This was beyond our control as we were only able to access the publicly available summary statistics for GLGC.

It is unlikely that this has affected the results because the overlap in causal genetic architecture was consistently high between European and Asian ancestry samples. Medication use should add to the error term and therefore could lead to a small downward bias. We have identified this as a limitation of our work (p9):

"As a limitation of our study, we did not adjust for use of lipid-lowering medication. This could in principle cause a small downward bias for the genetic effect estimates. However, few of the participants of the Ugandan and Chinese studies used lipid-lowering drugs. So this is unlikely to have an effect on the main conclusions of this work."

6) -The authors excluded all of chromosome 6 from genetic correlation analyses stating that LD calculations failed. How many known loci were excluded as a result? Did LD calculations fail for all ancestry-matched references, or just some? Did the authors try to exclude highly variable regions of chr 6 before excluding the whole chromosome?

This is actually a mistake in the methods description. We apologise and thank the reviewer for picking this up. We did not exclude the entire chromosome, just one region. The problems with chromosome 6 occurred when calculating the LD scores that are needed to estimate the trans-ethnic genetic correlations. LD across a very long range can interfere with that. Therefore, we excluded the MHC region from all analyses which resolved the estimation problems. We have now corrected this in the methods description (p12):

" The estimation of LD scores failed for chromosome 6 for some groups. We therefore left out the major histocompatibility complex (MHC) region (positions 28,477,797 to 33,448,354) from chromosome 6 from all comparisons."

7) -There is no description of genotyping or imputation (arrays used, QC procedures, exclusions, imputation methods, references used).

Basic quality control and imputation of the studies was not part of this work. These details have been described elsewhere. We had provided references. However, it was not clear from our description that they cover all of these aspects. We have now clarified this on page 11. Supplementary Table 6 (after renumbering) contained some information, such as the arrays. We have now added another table (new Supplementary Table 5) to include more details about the QC and imputation of each study.

Minor Concerns

8)-When reporting the results, the authors are often vague (e.g. "correlations ... were close to 1", "largely consistent correlation", "correlations were high"). It would help to report specific values and ranges within the text to assist the reader in following the interpretation and discussion of results.

We revised the manuscript and replaced these expressions or added numbers to support them (p 3, 5, 8, 9).

9)-The authors use the term causal, but they are not utilizing methods that determine causality. Instead of referring to causal variants, refer to associated loci, or tag/lead SNPs underlying loci.

The reviewer is right in pointing out that causality is very difficult to establish and that our work does not provide evidence for the causality of specific loci. At no place do we claim causality of an individual variant. Instead, our work refers to causality as a theoretical construct that plays an important role in our models. The key question we are trying to address is whether causal variants for lipids are shared across populations. It is already known that association results for associated loci or tag/lead SNPs are often inconsistent across populations. What we are trying to address is whether this is caused by lack of sharing of causal variants for lipids or by LD and frequency differences between populations resulting from divergent population histories. This would be easy to address if the causal variants were known. The diverse methods we apply are motivated by the fact that we do not know the causal variants and might never know it for many of the loci. The methods we used, such as trans-ethnic colocalization or genetic correlation, aim to answer the question without having to specify the causal variant. We have clarified that in the introduction (p4):

"Here we ask the fundamental question whether causal variants for blood lipids, a major cardiovascular risk factor, are shared across populations. Heterogeneity in effects of variants could result from epistasis or gene-environment interactions. However, the causal variants are usually unknown. The differences in LD structure between populations make it difficult to compare the observed associations between ancestry groups because the effect of a variant

depends on its correlation with the causal variant(s). Differences in frequency also impact on the power to detect associations in other ancestry groups.

We employed several strategies which account for these effects and do not require knowledge of the specific causal variants to quantify the extent to which genetic variants affecting lipid biomarkers are shared between individuals from Europe/North America, Asia, and Africa."

10)-Minor typos throughout the article (e.g. in Introduction, fifth line, "representation" should be "represent"; "und" on page 14, etc.)

We are grateful to the reviewer for spotting these typos. We have now thoroughly proofread the manuscript and corrected typos.

11)-The authors use the term replication, but generalization is more appropriate for the non-European descent populations.

We replaced "replication" with "reproducibility" or "transferability" throughout the manuscript.

12)-The authors mention overlapping confidence intervals in results, but only report SE in tables and do not supply any figures that illustrate this. While one can calculate the CI from the data provided, it would make it easier if these numbers were provided for easy assessment (i.e. in forest plots).

We have added confidence intervals to Table 2. The values are already illustrated in figure 2 and we feel another presentation of these findings could be repetitive.

13)-Section headers in the results would be beneficial.

We have added sub-headers to the results section.

14)-The authors state that they obtained publicly available BMI data from Pulit et al.; however, the work referenced was on WHRdjBMI. I believe they meant to cite Yengo et al. (PMID: 30124842).

We thank the reviewer for pointing this out. We cited this paper because the public download page for the summary statistics referred to Pulit et al (<https://zenodo.org/record/1251813#.XRY1oS2ZOV5>). In order to be certain, we contacted the lead authors of both papers. Unfortunately, we did not receive a reply. However, Yengo et al is the correct reference, so we have replaced the citation.

15)-Manhattan plots are not necessary.

We have removed the Manhattan plots.

16)-The authors adjust for principal components in the SNP associations, but do not mention them in the PRS analyses.

In general we used mixed models to account for relatedness and population structure for both the single SNP and the PRS association analyses. For APCDR-Uganda, UKHLS, and HELIC it was not necessary to adjust for PCs in these analyses as there was no evidence of unadjusted population stratification. We have added the QQ plots below for HDL (Figure R1). They were similar for other biomarkers. For UKHLS, the inflation estimates lambda were 1.002, 1.000, and 1.004 for HDL, LDL and TG, respectively. For Uganda, the inflation estimates lambda were 1.000, 1.004, and 1.005 for HDL, LDL and TG, respectively. For China Kadoorie Biobank (CKB) PCs were included in both single SNP and PRS association analyses to improve inflation. Recruitment for CKB occurred at 10 different rural and urban locations across China leading to increased population structure. The resulting inflation estimates lambda after PC adjustment were 1.063, 1.050, and 1.053 for HDL, LDL and TG, respectively.

Figure R1: QQ plots for associations with HDL for a) UKHLS, b) CKB, c) BBJ, d) APCDR-Uganda

17)-A brief description of what defines a credible set would be helpful in the Results in addition to the methods to assist in interpretation.

To clarify how we generated credible sets, we have added the definition we used to the results section (p 5):

"We defined the credible set as variants correlated at $r^2 > 0.6$ with the lead SNP from the European discovery study. Correlation was estimated from the 1000 Genomes Project samples with European ancestry."

18)-The authors do not justify the use of different transformation methods used across studies. Why not harmonize the phenotypes.

We have now harmonised phenotype transformations (see reply to Reviewer 3, comment 13)). The covariates we adjusted for were generally consistent between studies with the exception of inclusion of PCs for CKB (see reply to comment 16)).

19) -Why did the authors choose not to meta-analyze the HELIC studies? Is there prior evidence of population stratification?

These two groups are separate isolated populations from different regions in Greece without any notable recent intermixing between them (see Figure R4b below). They have been described in detail by Panoutsopoulou et al, 2014, in Nature Communications (doi: DOI: 10.1038/ncomms6345). Because we are studying differences between populations it did not appear appropriate to combine these two different populations.

20) -The equation for trans-ethnic colocalization is missing or did not transfer during pdf creation.

We assume that this must have been caused by the automatic reformatting or different computer platforms. We will be working with the Nature Communications team to make sure equation is visible in the revised version.

21) -The figure headings are not descriptive enough (i.e. off-diagonal elements are not defined in Figure 1, Table 4 should explain why so few loci are represented, Figure 2 is not very informative, etc.).

We have added more detail to the figure headings to make the display items self-explanatory. For example, the heading of Table 3 (formerly Table 4) now reads (p22):

"Table 4. Association of established lipid-associated loci with body mass index by whether the locus was transferable to APCDR-Uganda. Association results are based on $N \geq 484,680$ samples from the meta-analysis between GIANT and UK Biobank. Results are shown exclusively for loci where there was clear evidence for or against transferability to APCDR-Uganda (see Methods for more details)."

Reviewer #2 (Remarks to the Author):

This study investigated sharing genetic variants of major lipids profile across different ethnic groups by several strategies. First, they estimated replication rates of lipid biomarkers by selecting major lipid loci (with $p < 10^{-100}$) as credible set and defined replication with $p < 0.001$ in target cohorts. Secondly, they estimated genetic correlations between cohorts using trans-ethnic genetic correlation method for large studies and polygenic score association for smaller cohorts. Then they applied trans-ethnic colocalization to identify shared lipids loci by different ethnic groups. The aims of the study are clear and the paper is well organized. However, the following points should be addressed.

We thank the reviewer for this accurate summary of our work.

Major comments:

1) The interesting results presented may indicate that the genetic components of TG in African Americans might be distinct but major genetic variants of LDL and HDL share in some degree across different ethnic groups. However, the results were not discussed for the possible variation of lipids profile in this population. The lipids values are on average lower comparing with other populations (Supplementary Table 4). What is the phenotypic correlation between TG and HDL? The correlation is probably low according to the results in Supplementary Table 2. If it is true, any mechanism might affect shared genetic components of TG should be investigated or discussed.

Firstly, we would like to clarify that we did not use data from African Americans. The samples with African ancestry were recruited in rural Uganda in Africa.

We agree with the reviewer that our results suggest a lower degree of shared genetic factors for triglycerides between Ugandans and samples with European ancestry. This might indeed relate to different lipid profiles. The reviewer is correct in pointing out that the lipid profiles differed between the sample sets. However, the phenotypic differences should be a consequence rather than cause of environmental and genetic factors that differ between these populations. Generally, it is difficult to predict how differences in overall lipid levels relate to the sharing of genetic factors. For example, the average level of LDL is notably lower in APCDR-Uganda than in the other studies. Nevertheless, many of the established LDL loci from European discovery studies also affect LDL levels in Uganda. The same does not seem to apply to triglycerides.

It is an interesting question whether there might be differences in the inter-relationships between different lipid biomarkers and how they link to genetics. While a comprehensive investigation of this question is beyond the scope of this work, we tried to provide some statistics based on our data.

[Redacted]

Table R2: GLGC2017 p-values for the associations with HDL, LDL and TG for established lipid loci that did or did not demonstrate evidence of transferability to APCDR-Uganda

Transferable	Trait	rs-id	Chr	Position	Annotation	P-HDL	P-LDL	P-TG
no	HDL	rs11755393	6	34824636	UHRF1BP1	4.20E-23	0.5	0.46
no	HDL	rs1178979	7	72856430	BAZ1B	1.30E-26	0.00058	1.50E-179
no	HDL	rs4731702	7	130433384	KLF14	1.20E-35	0.016	1.10E-23
no	HDL	rs2954033	8	126493746	NSMCE2	3.00E-61	4.00E-33	1.80E-176
no	LDL	rs4245791	2	44074431	ABCG8	0.023	1.70E-120	3.40E-10
no	LDL	rs3846662	5	74651084	HMGCR	0.023	3.30E-128	0.15
no	LDL	rs2737229	8	116648565	TRPS1	0.64	8.90E-15	7.10E-06
no	LDL	rs635634	9	136155000	IL6R	9.20E-07	4.90E-109	0.0012
no	LDL	rs2000999	16	72108093	HPR	0.039	4.00E-71	7.70E-10
no	TG	rs1260326	2	27730940	GCKR	0.031	7.80E-17	0
no	TG	rs2943641	2	227093745	IRS1	7.80E-41	0.018	4.90E-33
no	TG	rs6905288	6	43758873	VEGFA	5.40E-21	0.00036	9.00E-35
no	TG	rs11820589	11	116633862	APOA5	4.00E-28	1.40E-05	4.40E-133
no	TG	rs58542926	19	19379549	TM6SF2	0.0021	6.5E-93	3.7E-125
yes	HDL	rs643531	9	15296034	TTC39B	3.80E-42	0.31	1.90E-06
yes	HDL	rs1800588	15	58723675	LIPC	0	0.76	8.60E-54
yes	HDL	rs3764261	16	56993324	CETP	0	1.50E-29	1.80E-36
yes	HDL	rs16942887	16	67928042	PSKH1	9.80E-93	0.76	7.30E-05
yes	LDL	rs12740374	1	109817590	CELSR2	1.50E-47	0	6.20E-07
yes	LDL	rs1367117	2	21263900	APOB	5.20E-12	3.60E-278	2.30E-15
yes	LDL	rs6511720	19	11202306	LDLR	4.3E-09	0	0.13

2) It is confusing that the authors used different datasets/cohorts for different strategies. The purpose of the paper is to quantify genetic sharing among populations from Europe/North America, Asia and Africa. As the authors mentioned, they have both phenotype and genotype data available from eMERGE, UKHLS, CKB, HELIC, and APCDR-Uganda and summary statistics from GLGC and BBJ. The authors tested replication rates

with all cohorts except for HELIC. For genetic correlation estimation and trans-ethnic colocalization, the authors excluded eMERGE. Explanations are needed for excluding specific cohorts from different analysis purposes.

We appreciate that diverse data sets were used without sufficient explanation why some data sets were excluded from certain analyses. Please see our replies to comments 2) and 4) by Reviewer 1.

Additionally, we excluded HELIC from the replication analyses (now renamed reproducibility) because of the relatively small sample size. In these samples it would not be possible to ascertain whether absence of reproducibility is due to lack of power or lack of generalisation. We have now clarified that in the manuscript (p12):

" These analyses excluded the HELIC studies because the smaller sample size makes it difficult to differentiate between lack of power and lack of reproducibility."

3) According to the sample size limitation, the authors can only estimate trans-ethnic genetic correlations for some large cohorts. The polygenic scores were alternatively used to test sharing genetic correlations among smaller ethnic groups. In order to have better understanding and comparison for genetic correlations, it will be helpful to include all available cohorts for the associations with polygenic scores.

We have now carried out the GRS analyses for all samples where raw genotype and phenotype data were available. Please see our response to comments 2) by Reviewer 1.

4) The novel part of this study was to use the trans-ethnic colocalization to identify shared loci accounting for LD structure in different ethnic groups. However, they need a clear description and consistency of the results. For example, the p values in Figure 3 are not consistent with those in Supplementary Table 3. The Result in Page 6, Paragraph 3, Line 4 indicates LPL loci but shows ABC1A loci in Figure 3c. More explanations of results from trans-ethnic colocalization, ie, Figure 3a and 3d, will be necessary. It seems that the regional plots of Japan and China show a similar pattern but the p values are significant different.

We thank the reviewer for pointing this out. The inconsistencies between Supplementary Table 3 (now 4) and Figure 3 (now 4) result because colocalization can be carried out using either of the two data sets that are being compared as the reference. We appreciate that this was not sufficiently clear from the manuscript and have decided to simplify the results by using UKHLS as the reference set for all comparisons so that the figure and table are now showing consistent results. We have also simplified Figure 3 and now present only one locus as an example (see also our reply to Reviewer 1 comment 3). Following the reviewer's suggestion, we have also provided additional explanation of this figure in the results section (p7):

" Figure 4 shows the regional association plots of this locus for each data set as an example to demonstrate that differences in LD and frequencies lead to different association patterns. As colocalization can account for such differences, the result from the analysis comparing the European and Asian studies was nevertheless statistically significant ($p < 0.001$)."

5) The definition of polygenic scores and genetic risk scores is sometimes confusing. According to the paper by Khera et al (Nature Genetics 2018), it is more suitable to refer polygenic scores using whole genome data and genetic risk scores by GWAS significant SNPs.

We have changed our notation to agree with Khera et al and use genetic risk scores throughout the manuscript now.

6) The authors used eMERGE data but didn't mention it in the Introduction.

Please see our reply to comments 4) by Reviewer 1.

7) Because of the multiple testing issue, what is the evidence for the significant threshold definition, ie, $p < 0.001$ for replication, and $p < 0.05$ for trans-ethnic colocalization?

We appreciate that this is an important issue, especially as different thresholds were used for different analyses.

Reproducibility of established lipid loci

The challenge is that the causal variant is unknown. So we tested multiple variants correlated with the lead SNP, the credible set, which can increase the risk for false positives. Therefore, we established a p-value threshold by random sampling of SNPs windows across the genome and assessing their minimum p-value in each study. This is described on p12.

"We used a p-value threshold of $p < 10^{-3}$. This threshold was derived by computing the minimum p-value in 1000 random windows of 50Kb for each study. Less than 5% of random windows had a minimum $p < 10^{-3}$ for the non-European ancestry studies. While this p-value threshold might not be appropriate to provide conclusive evidence of reproducibility for individual loci, we used this to test evidence of reproducibility across sets of loci."

Trans-ethnic genetic correlations

The 95% confidence intervals for each estimate overlapped with 1, suggesting no evidence for deviations from shared causal genetic architecture. A more stringent threshold would not change this finding. For the cross-biomarker tests a multiple testing adjustment similar to the GRS (below) is appropriate given that we also consider combinations of different lipids with each other across studies. We have now clarified the approach on p13.

"We used a Bonferroni correction to adjust for multiple testing of three traits with each other ($p < 0.05/9 = 0.0056$)."

Genetic risk scores

We described our approach on p14:

"We used a Bonferroni correction to adjust for multiple testing of three PRS with three different biomarker outcomes ($p < 0.05/9 = 0.0056$)."

Trans-ethnic colocalization

We use a p-value threshold of 5% to declare significant colocalization. We justified this on p15:

"We used a significance threshold of $p < 0.05$ given the evidence of association of the established lipid loci in Europeans and the overall evidence for shared causal genetic architecture across populations for most lipid traits from our other analyses."

Minor comments:

Table 1: please add the numbers of SNPs with $p < 10^{-100}$ and $p > 10^{-100}$, respectively.

We have added this information to Table 1 (p24).

Please use consistent names for all cohorts. For example, I assume 'UG' in Table 1 should be 'APCDR-Uganda'. The Figure 3 shows each cohort with the country name.

We thank the reviewer for pointing this out. We have corrected this and now use consistent study names throughout the manuscript.

In Page 10, Paragraph 1, Line 6, the study details in 'Supplementary Table 3' should refer to 'Supplementary Table 4'.

We thank the reviewer for pointing this out. We have corrected this.

Reviewer #3 (Remarks to the Author):

Telkar et al. present an interesting and timely study on the degree with which the genetic architecture of lipid levels is shared between populations of different ancestry. The paper is well written, and relies on impressive amounts of datasets and statistical analyses. It uses state-of-the-art software and propose a modified version of JLIM for identifying shared genetic effects across populations with different ancestries. However, the datasets and analyses used in the paper are in my opinion not always described with sufficient detail. I understand that this is difficult because of the large number of different analyses and datasets used in the paper, but they are necessary for me to review the study and for anyone wanting to replicate it. Indeed, the devil is in the details. Also, although the extension of JLIM to detecting shared genetic effects between ancestries is very interesting, and I would highly encourage the authors to pursue these ideas further, I am not convinced it is working as intended based on the simulations. However, the JLIM results are in my opinion just a minor point in this study, and I think it is still interesting as it indicates that the genetic architecture of lipid levels is somewhat consistent between populations.

We thank the reviewer and very much appreciate this encouraging feedback.

Comments:

1) - There's no description of how you processed the APCDR-Uganda, eMERGE, and CKB data, which I presume you performed some QC on. E.g., did you filter out individuals of non-Asian ancestry in the Asian cohorts? Similarly, did you filter out individuals of with non-European ancestry in the European eMERGE, etc. A PCA plot for these cohorts, e.g. a projection to the PCA plot from 1000 genomes samples would clearly reveal any admixed individuals. Did you filter out SNPs, or perform relatedness filtering on the samples? All of this information is important to understand how to interpret the results. It would allow you to rule out any possible association signals due to admixed individuals. If this information is left out here because the data was already processed and QC'd in another study, I would appreciate if the specific citations in question were clearer. Similarly, I could not find information on how you QC or select British individuals from the UKB. I am sure this is all trivial or standard, but I would appreciate some descriptions. Finally, regarding the HELIC and UKHLS datasets, (although perhaps not necessary) I would appreciate if you verified the ancestry filtering in these, e.g. by projecting them onto a 1000 genomes PCA plot.

Please see our response to Reviewer 1 point 7). Details of the genotyping and quality control for each study have been reported elsewhere which we have now clarified. We have also added an additional Supplementary Table 5 with details of the genotyping, imputation and QC for each study.

For the simulations, for UK Biobank we applied the QC and ancestry selection provided with the official data release (Bycroft et al., 2018, *Nature Genetics*) and then selected a random subset of 50,000 individuals. We have now clarified this in the manuscript (p15).

Regarding the ancestry assessment, for each study ancestry was validated genetically through PCA (see Figures R2-5 below). Individuals with admixed ancestry or ancestries not matching the majority of the samples in that study were excluded.

Figure R2: The first two principle components from a principle component analysis (PCA) for all samples genotyped for the UK Health and Longitudinal Study (UKHLS). Of these, 340

samples with non-European ancestry seen to deviate from the European cluster were excluded from our analyses.

[Redacted]

Figure R4: **(a)** MDS analysis of HELIC-MANOLIS, HELIC-Pomak and TEENAGE combined with populations from the 1000 Genomes Project. **(b)** MDS analysis of HELIC-MANOLIS villages, HELIC-Pomak villages and TEENAGE. Individuals from the MANOLIS cohort are depicted by the differently coloured hollow triangles with each colour corresponding to the village of origin. Individuals from the Pomak villages are depicted by the differently coloured hollow circles with each colour corresponding to the village of origin. The black solid circles depict individuals from TEENAGE representing the general Greek population. (from Panoutsopoulou et al, 2014, *Nature Communications*, <https://www.nature.com/articles/ncomms6345/figures/1>)

[Redacted]

2) - Is there any chance of sample overlap between these studies and the GLGC2017 study? Please clarify. If there is overlap, then the correlations will be (partially) driven by overfitting.

For the genetic correlations we used GLGC2013 because of the higher SNP density. All samples used to generate the GLGC results had confirmed European ancestry (Willer et al, 2013, *Nature Genetics*). Participants for the other studies were recruited in Uganda, China and Greece and genetically each group matched the expected ancestry (see reply to comment 1). Therefore, it can be ruled out that there is sample overlap between the studies. We have added a clarification to this effect on p11:

"Each study confirmed sample ethnicity through PCA which rules out sample overlap between studies."

3) - Regarding JLIM simulations, how about using the same simulation setup to test for trans-ethnic colocalizations in a European dataset. This would represent the best possible case, a scenario, representing an upper limit on your statistical power. Indeed, given the huge simulated genetic effect (phenotypic variance explained is likely between 1-20%, depending on MAF), the statistical power does not depend on the sample size differences between samples (as it's probably near maximum) but rather on LD differences. In which

case, I am worried that the model may not be able to detect shared associations even if they are true (due to LD), leading to downwards biased colocalization frequency estimates. Perhaps you could address this in the discussion, or better yet, extend simulations to study this further (although that's possibly better suited for future studies where you also study the effect of multiple loci, etc.).

We would like to thank the reviewer for these suggestions. We agree that a matched ancestry set to derive an upper bound for the best possible performance helps clarify the power losses through genetic distance. We therefore also compared UK Biobank as a reference with UKHLS. The power was 0.89. This is comparable to the power seen for comparison between UK Biobank and the Chinese data set (0.93). The type I error was close to 5% (0.0505).

Regarding the simulated effect size, we would like to point out that a beta of 0.25 is not unrealistic. Table R3 shows an excerpt from table from GLGC2017 (Liu et al, 2017, *Nature Genetics*) showing the results for the 54 SNPs with the smallest p-values for lipid associations. The average (standardised) effect size in this set was 0.21. The largest effect size of any common variant is 0.54. Additionally, we have chosen $\sigma = 1$ to match the traits variance for standardised phenotypes.

Table R3: Excerpt from Supplementary Table 12 from GLGC2017 (Liu et al, 2017, *Nature Genetics*) showing the lipid association results in GLGC for the 54 SNPs with the smallest p-values for lipid associations.

POSITION	RS	REF/ALT	ANNOTATION	N	ALT FREQ	Trait	p-value	Beta	SE
1:109817590	rs12740374	G/T	CELSR2:Utr3	294565	0.22	LDL	0	-0.16	0.0032
1:109818530	rs646776	C/T	CELSR2:Intergenic	264187	0.77	LDL	0	0.16	0.0033
1:55505647	rs11591147	G/T	PCSK9:Arg46Leu	265213	0.015	LDL	0	-0.48	0.011
1:63118196	rs10889353	A/C	DOCK7:Intron	304422	0.33	TG	6.4E-170	-0.077	0.0028
11:116586283	rs7350481	T/C	C1orf158:Intergenic	303685	0.91	TG	0	-0.23	0.0047
11:116633862	rs11820589	G/A	BUD13:Pro148Leu	138265	0.066	TG	4.4E-133	0.19	0.0079
11:116639104	rs10790162	A/G	BUD13:Intron	297913	0.92	TG	0	-0.26	0.0047
11:116648917	rs964184	G/C	PRAMEF2:Intergenic	305699	0.85	TG	0	-0.25	0.0036
11:116662407	rs3135506	G/C	APOA5:Ser19Trp	305699	0.059	TG	0	0.24	0.0055
11:116701354	rs138326449	G/A	APOC3:Essential_Splice_Site	137279	0.0018	TG	3.6E-138	-1.14	0.046
11:116722041	rs10047462	G/T	SIK3:Intron	299927	0.86	TG	9.9E-180	-0.11	0.0039
15:58678512	rs10468017	C/T	SNX27:Intergenic	316391	0.27	HDL	1.8E-306	0.11	0.0029
15:58683366	rs1532085	A/G	SNX27:Intergenic	316391	0.59	HDL	1.2E-295	-0.096	0.0026
15:58723675	rs1800588	C/T	SNX27:Intergenic	316391	0.24	HDL	0	0.12	0.003
16:56988044	rs173539	C/T	OR10K2:Intergenic	292803	0.32	HDL	0	0.23	0.0028
16:56989590	rs247616	C/T	OR10K2:Intergenic	316391	0.31	HDL	0	0.24	0.0028
16:56993324	rs3764261	C/A	OR10K2:Intergenic	310132	0.31	HDL	0	0.24	0.0028
16:56995935	rs34065661	C/G	CETP:Ala15Gly	275159	0.0048	HDL	5.6E-103	0.43	0.02
16:57002732	rs9939224	T/G	CETP:Intron	316391	0.79	HDL	0	0.2	0.0031
16:57006590	rs7499892	C/T	CETP:Intron	296037	0.19	HDL	0	-0.23	0.0034
16:57016092	rs5882	G/A	CETP:Val422Ile	295475	0.65	HDL	5.5E-241	-0.092	0.0028
17:41926126	rs72836561	C/T	CD300LG:Arg82Cys	313322	0.028	HDL	8.1E-111	-0.17	0.0077
18:47160953	rs7241918	G/T	MTHFR:Intergenic	310567	0.85	HDL	1.2E-104	0.077	0.0036
19:11202306	rs6511720	G/T	LDLR:Intron	295826	0.11	LDL	0	-0.21	0.0043

19:19379549	rs58542926	C/T	TM6SF2:Glu167Lys	319677	0.074	TC	7E-155	-0.13	0.0049
19:45316588	rs28399654	G/A	BCAM:Val196Ile	274122	0.027	LDL	7.5E-232	-0.27	0.0084
19:45395266	rs157580	G/A	TOMM40:Intron	319677	0.63	TC	3.7E-168	0.073	0.0026
19:45410002	rs769449	G/A	APOE:Intron	293853	0.11	LDL	0	0.19	0.0042
19:45412079	rs7412	C/T	APOE:Arg176Cys	183156	0.075	LDL	0	-0.54	0.0064
19:45414451	rs439401	T/C	ATP13A2:Intergenic	305699	0.63	TG	2.7E-168	0.075	0.0027
19:45415640	rs445925	G/A	ATP13A2:Intergenic	290263	0.11	LDL	0	-0.32	0.0043
19:8429323	rs116843064	G/A	ANGPTL4:Glu40Lys	280371	0.02	TG	4.2E-175	-0.27	0.0097
2:21229160	rs5742904	C/T	APOB:Arg3527Gln	281793	0.00039	LDL	1.1E-104	1.48	0.068
2:21231524	rs676210	G/A	APOB:Pro2739Leu	305699	0.26	TG	4.9E-118	-0.071	0.0031
2:21263900	rs1367117	G/A	APOB:Thr98Ile	319677	0.29	TC	5.6E-231	0.092	0.0028
2:21294975	rs541041	G/A	TNFSF4:Intergenic	295826	0.81	LDL	1.3E-287	0.12	0.0034
2:27730940	rs1260326	T/C	GCKR:Leu446Pro	305699	0.63	TG	0	-0.12	0.0027
2:27748624	rs1260333	A/G	PTPN14:Intergenic	120928	0.57	TG	2.1E-114	-0.095	0.0042
2:44074431	rs4245791	C/T	ABCG8:Intron	270962	0.72	LDL	1.7E-120	-0.072	0.0031
5:74648603	rs12654264	A/T	HMGCR:Intron	295826	0.4	LDL	2.7E-128	0.066	0.0027
5:74651084	rs3846662	A/G	HMGCR:Intron	295826	0.48	LDL	3.3E-128	0.065	0.0027
7:72856269	rs2240466	G/A	BAZ1B:Intron	277144	0.11	TG	1.9E-165	-0.12	0.0044
7:72856430	rs1178979	T/C	BAZ1B:Intron	305699	0.18	TG	1.5E-179	-0.096	0.0034
7:73012042	rs35332062	G/A	MLXIPL:Ala358Val	305699	0.12	TG	5.2E-205	-0.12	0.0041
8:126507389	rs2954038	C/A	AMPD1:Intergenic	305699	0.72	TG	5.6E-197	-0.087	0.0029
8:19813529	rs268	A/G	LPL:Asn318Ser	301289	0.017	HDL	6.9E-154	-0.26	0.0099
8:19816934	rs301	T/C	LPL:Intron	305699	0.24	TG	0	-0.12	0.0031
8:19819439	rs326	A/G	LPL:Intron	316391	0.3	HDL	0	0.11	0.0028
8:19819724	rs328	C/G	LPL:Ser474Stp	305699	0.098	TG	0	-0.18	0.0044
8:19824492	rs13702	T/C	LPL:Utr3	305699	0.3	TG	0	-0.12	0.0029
8:19830921	rs10096633	C/T	GATAD2B:Intergenic	305699	0.14	TG	0	-0.15	0.0038
8:9183596	rs4841132	A/G	NECAP2:Intergenic	316391	0.9	HDL	1E-123	0.1	0.0044
9:107664301	rs1883025	C/T	ABCA1:Intron	316391	0.26	HDL	2.1E-118	-0.067	0.0029
9:136155000	rs635634	C/T	IL6R:Intergenic	276356	0.19	LDL	4.9E-109	0.077	0.0035

However, the reviewer is right that a range of effects were observed. We therefore repeated the simulations for all studies using betas of 0.10, 0.15, and 0.2. The type I error rates were generally close to 0.05 (new Supplementary Table 3). As seen in the new Figure 3, power decreased for smaller betas, in particular for the two smaller studies. Additionally, the power for APCDR-Uganda was consistently less than the power of the other studies. The power for the full set of 72,000 CKB samples was consistently above that for the ancestry matched simulation using UKHLS (N=9,000). The updates results section now read (p6f):

"In order to assess its performance for GWAS results from samples with different ancestry, we carried out a simulation study. UK Biobank (UKB) was used as a European ancestry reference and compared to CKB and APCDR-Uganda. In order to derive an upper limit for the power, we compared UKB to the ancestry-matched UKHLS set. Phenotypes were simulated. Effect size estimates were varied between 0.10 and 0.25 in order to represent a range similar to that observed for the major lipid loci⁴. In the simulations of distinct causal variants in the non-European and the reference group, the frequencies of false positives were as expected close to 0.05 (Supplementary Table 3, Supplementary Figure 1). The power to detect shared associations for betas of 0.25 was 73.1% for APCDR-Uganda, 93.1% for CKB and 0.89 for UKHLS (Figure 3). To investigate whether the lower power for APCDR-Uganda could be due to its smaller sample size, we reran the analyses for CKB using a random subset of samples matching the sample size of APCDR-Uganda. For effect sizes less than 0.2, the results from this analysis revealed decreased detection power relative to the full CKB set but still consistently higher than APCDR-Uganda. This suggests that the power of this trans-ethnic colocalization method decreases somewhat with greater genetic distance between the populations that are compared. "

The decreased performance for APCDR-Uganda for smaller betas represents a limitation of the current implementation which we now discuss in the manuscript (p 8):

"Simulations showed consistent control of type I error rates as well as power greater than 80% to detect shared associations between samples with European and Chinese ancestry for SNP effects greater or equal to 0.15. However, power was decreased for comparisons between samples from APCDR-Uganda and UK Biobank (51.5-73.1%). Hence, for the current implementation non-significant colocalization should not be considered as definitive evidence for the absence of shared causal variants when comparing African and European samples. Future work should address this through better modelling of the LD structure. Moreover, for many of the major lipid loci, more than one independent association signal was identified in discovery GWASs¹⁵. When these are located in close proximity to each other, they can interfere with the trans-ethnic colocalization analysis because JLIM assumes a single causal variant. Therefore, future work should extend this approach to accommodate loci harbouring multiple causal variants."

Overall, we agree with the reviewer that this new method requires further work including improved handling of LD, accounting for multiple causal variants in a region and further simulation work to assess the performance. We aim to implement these in the future. However, as the reviewer points out, this is beyond the scope of this already very content-rich manuscript. We hope that the additional simulations and added note of caution are satisfactory.

New manuscript Figure 3. Power of trans-ethnic colocalization to detect shared associations for different effect sizes (Beta). Based on a simulation study comparing 50,000 samples with European ancestry to UKHLS, APCDR-Uganda ("UG"), and China Kadoorie Biobank ("CKB"). Additionally, China Kadoorie Biobank was downsampled to N=4597 ("CKB_4K") to match APCDR-Uganda.

4) - p. 13, The modified version of JLIM uses LD scores that are estimated using ancestry matched samples from the 1000 Genomes Project v3. However, it is unclear to me how you actually do this. Did you match the ratio of ancestry in your data, say 50% EUR, 20% Uganda and 30% Chinese with similar ratios in the 1000 genomes data, and use that to estimate LD? Which 1000 genomes population did you use as representatives? I would prefer more details here.

Using LD estimates from external reference samples makes it possible to use summary statistics for one of the samples, e.g. BBJ for which raw data were not available. For the second study, LD is computed directly from the raw data. Therefore, the external LD scores need to match the ancestry of the sample with summary statistics. For example, we selected the 1000 Genomes Project samples with Japanese ancestry for BBJ. We used a modified version of this script <https://github.com/cotsapaslab/jlim/blob/master/bin/fetch.refld0.EUR.pl> provided by Chun et al (2017, *Nature Genetics*) to compute the LD scores for the ancestry-matched samples specifically for the region of interest. We have provided a more detailed description of this in the methods section on page 15:

"For the reference sample set, it was possible to use genome-wide summary statistics for the analysis. For this set, LD scores were estimated using a subset of samples from the 1000 Genomes Project v3 that had matching ancestry to that study. The second sample set needed raw genotype data and LD was estimated directly for these samples."

Minor comments:

5)- On p. 5, “biomarkers” probably refers to lipids. Is that true? This is possibly confusing.

Biomarker indeed referred to the blood lipid biomarkers. We have replaced biomarker throughout the manuscript with lipid biomarker to avoid confusion.

6)- On p. 5, you “estimated” the PRS associations in HELIC, APCDR-Uganda and also - UKHLS. Instead of “estimated” would prefer validated, or predicted to somehow make clear the distinction between training and validation (which I assume this is).

We thank the reviewer for pointing this out! We have replaced "estimated" with "validated" (now p6).

7)- Regarding supplementary figure 1, how about using a QQ-plot instead, and possibly summarizing in a mean/median p-value. You can also test for significance using a Kolmogorov-Smirnov test.

Following the reviewer's suggestion, we have replaced Supplementary Figure 1 with QQ plots for trans-ethnic colocalization p-values based on simulations with distinct causal variants in the two sample sets. The plots show well-controlled type I errors.

8)- On p. 7, first paragraph in Discussion. I suggest to drop the parenthesis around “triglyceride loci except in APCDR-Uganda”, because this is an important result.

We have followed the reviewers suggestion and removed the parenthesis.

9)- On p. 10. Why does eMERGE suddenly show up in the Methods section, when it's not even mentioned in the results or introduction?

Please see our reply to Reviewer 1 comment 4). We have now removed eMERGE.

10)- p. 10, first paragraph. You probably mean supplementary table 4, not 3.

We thank the Reviewer for pointing this out. We have now corrected this.

11)- Why use GEMMA for GWAS in APCDR-Uganda and BOLT-LMM for eMERGE? I would prefer greater consistency in your choices of methods and analyses.

Please see our reply to Reviewer 1 comment 4). We have now removed eMERGE.

12)- In the GWAS, did you add some covariates to the analyses, e.g. PCs, sex, age? Please avoid heritable covariates (see Aschard et al., AJHG 2015).

We adjusted for age and gender and, for CKB, for regions and PCs as this was necessary to control for population structure.

13)- I would appreciate if you clarified what you mean by “rank-based inverse normal transformation”, e.g. using an equation or two. I assume it means you ranked the lipid

levels, and assumed the original values followed a Gaussian distribution, and then obtained the corresponding z-scores. Also, why do you later use a Box-Cox transformation? Again, I would recommend greater consistency in choices of analyses methods.

The reviewer's description of rank-based inverse normal transformation is correct. We have now added more detail on page 13.

"To ensure values are normally distributed, we used rank-based inverse normal transformation for all biomarkers and data sets which involves ordering values first and then assigning them to expected normal values."

In order to be consistent, we have now used rank-based inverse normal transformation for all data sets. The changes were marginal (maximum changes of .005 to effect estimates) (Table R4 below).

Table R4: Original PRS associations and updated results after using rank-based inverse normal transformation also for UKHLS and HELIC

Trait	N	Original values			With rank-based inverse normal transformation		
		Correlation (SE*)	Confidence interval	P-value	Correlation (SE*)	Confidence interval	P-value
UKHLS							
HDL	9706	0.284 (0.010)	0.264, 0.304	8.34x10 ⁻¹⁶⁵	0.285 (0.010)	0.265, 0.305	4.52x10 ⁻¹⁶⁶
LDL	9767	0.273 (0.010)	0.253, 0.293	8.38x10 ⁻¹⁵⁵	0.274 (0.010)	0.254, 0.294	1.32x10 ⁻¹⁵⁵
Triglycerides	9635	0.203 (0.010)	0.183, 0.223	2.62x10 ⁻⁸⁶	0.204 (0.010)	0.183, 0.223	9.62x10 ⁻⁸⁷
HELIC-MANOLIS							
HDL	1186	0.276 (0.029)	0.219, 0.333	8.65x10 ⁻²⁰	0.279 (0.029)	0.222, 0.336	4.08x10 ⁻²⁰
LDL	1186	0.230 (0.029)	0.173, 0.287	1.89x10 ⁻¹⁴	0.229 (0.029)	0.172, 0.286	2.41x10 ⁻¹⁴
Triglycerides	1176	0.237 (0.030)	0.178, 0.296	3.01x10 ⁻¹⁴	0.235 (0.030)	0.176, 0.294	4.52x10 ⁻¹⁴
HELIC-Pomak							
HDL	1078	0.272 (0.030)	0.213, 0.331	9.67x10 ⁻¹⁸	0.268 (0.030)	0.209, 0.327	2.39x10 ⁻¹⁷
LDL	1075	0.285 (0.030)	0.226, 0.344	1.35x10 ⁻¹⁸	0.290 (0.030)	0.231, 0.349	3.04x10 ⁻¹⁹
Triglycerides	1066	0.235 (0.030)	0.176, 0.294	1.68x10 ⁻¹³	0.234 (0.030)	0.175, 0.293	2.00x10 ⁻¹³
APCDR-Uganda							
HDL	6407	0.121 (0.012)	0.098, 0.145	6.06x10 ⁻²²			
LDL	6407	0.280 (0.012)	0.257, 0.304	1.91x10 ⁻¹⁰⁷			
Triglycerides	6407	0.063 (0.013)	0.038, 0.089	4.46x10 ⁻⁷			
CKB							
HDL	20810	0.180 (0.018)	0.145, 0.215	1.4x10 ⁻²²			
LDL	17662	0.198 (0.019)	0.161, 0.235	3.2x10 ⁻²⁶			

Triglycerides	20222	0.139 (0.020)	0.100, 0.178	3.8×10^{-12}
---------------	-------	---------------	--------------	-----------------------

14)- On p. 12. Why was it not possible to compute transethnic genetic correlations for the UKHLS, the HELIC cohorts, and APCDR-Uganda data? I would appreciate more details on this.

The Popcorn algorithm did not achieve convergence or returned heritability of 0 when we ran it for these data sets. Genetic correlations using Popcorn can only be computed for traits with non-zero heritability. The SNP heritability has previously been estimated for Uganda and UKHLS with different methods and was significantly different from 0 for all three lipid biomarkers (Heckerman et al, 2016, *PNAS*; Prins et al, 2017, *Scientific Reports*). So the Popcorn results were incorrect. We attempted multiple modifications, including different SNP filtering and different combinations of data sets. We also reached out to the lead author of this method (Brielin Brown) and he was not able to resolve this either. These problems occurred for the smaller sample sets while there were no problems for any of the larger ones. The requirement for large sample sizes for this method was already demonstrated in the original publication (Brown et al, 2016, *AJHG*) albeit without specifying a threshold.

15)- When calculating PRS, I wonder how the r^2 threshold affects predictions when predicting into individuals of different ancestry than the training data had. Perhaps predictions improve if you increased the threshold to 0.2 (or larger values)?

Including correlated SNPs in the PRS violates the assumptions of the model because those SNPs capture partially the same information. In principle, this should decrease the predictive accuracy of the score. Correlated SNPs can be accommodated by using SNP weights that are adjusted for their LD. For example, regression coefficients from a joint regression including both SNPs from the pair rather than from single SNP regressions. In practice, this is often not possible because only summary statistics from single SNP regressions are made available for the training sets, as was the case here. We were unable to gain access to raw data or more detailed analyses for GLGC. Moreover, in the case of different ancestry groups, the different LD patterns make it more challenging to account for correlations. To address this, we removed one from each pair in LD based on any of the data sets. We also restricted to the final SNP set for the PRS to those that were in linkage equilibrium, common and well imputed across all data sets. This ensured comparability of findings across sample sets. Otherwise better PRS performance could be driven by larger numbers of SNPs incorporated for one ancestry group but not another.

The reviewer might be suggesting that by including correlated variants the likelihood might increase that one of them is also correlated with the causal SNP in the other ancestry groups. This could improve prediction. It is not clear how it would balance with the negative impact of including multiple variants for the same association signal with unadjusted weights and could be affected by a number of factors. In our view this question requires in depth work on assessing strategies to improve trans-ancestry PRS performance. This is beyond the scope of our project where the focus is on assessing transferability and comparing this across populations.

16)- I liked the fact that you used mixed linear models to account for relatedness in the PRS correlations. Would it make sense to add a PCs as covariates to the model as well, for

robustness (in case there are environmental factors that correlate with PCs)? I'm also curious, whether one would see any association with PCs (or 1000 genomes PCs projections) in these populations?

We did not include PCs as covariates because in the genome-wide SNP association analysis there was no evidence of inflation suggesting that population stratification was well controlled. The exception was CKB where adjusting for PCs improved inflation in that context (see response to Reviewer 1, comment 16)). Therefore, PCs were included in the newly added GRS analyses for CKB.

As a sensitivity analysis we tested whether inclusion of PCs affects the GRS association results for UKHLS and CKB (Table R4 below). The effect estimates for GRSs were highly consistent between the models with and without PCs as covariates. The PCs were generally not associated with lipids in UKHLS but one of the PCs was associated in CKB. For HDL and TG the p-values for all PCs were > 0.05 in UKHLS. For LDL, PC7 had p=0.013. However, after Bonferroni correction for 10 PCs this is not significant. For CKB, PC9 was associated with HDL with $p=4.4 \times 10^{-5}$. This might be explained by greater dietary/lifestyle differences between different geographic locations in China. These sensitivity analyses were based on a linear model in R and excluded relatives because the mixed model implementations used for the main analyses do not return association results for covariates. Therefore, there are some differences in estimates to the main results for the GRS reported in the papers. However, the patterns are highly consistent.

Table R5: GRS associations with their target lipid biomarkers in UKHLS and CKB using a linear model adjusting for age, sex, and either PCs or not

Trait	N	with PCs		without PCs	
		Correlation (SE*)	P-value	Correlation (SE*)	P-value
UKHLS					
HDL	9185	0.262 (0.010)	$<2 \times 10^{-16}$	0.263 (0.010)	$<2 \times 10^{-16}$
LDL	9185	0.256 (0.010)	$<2 \times 10^{-16}$	0.256 (0.010)	$<2 \times 10^{-16}$
Triglycerides	9185	0.187 (0.010)	$<2 \times 10^{-16}$	0.188 (0.010)	$<2 \times 10^{-16}$
CKB					
HDL	13008	0.180 (0.008)	$<2 \times 10^{-16}$	0.179 (0.008)	$<2 \times 10^{-16}$
LDL	11091	0.178 (0.008)	$<2 \times 10^{-16}$	0.178 (0.008)	$<2 \times 10^{-16}$
Triglycerides	12726	0.126 (0.009)	$<2 \times 10^{-16}$	0.126 (0.009)	$<2 \times 10^{-16}$

17)- p. 13, second paragraph, you mention that you used summary statistics, possibly implying that you also run it on samples with individual-level genotypes? I think the formulation could be improved.

We have clarified this point. The section now reads (p15):

"For the reference sample set, it was possible to use genome-wide summary statistics for the analysis. For this set, LD scores were estimated using a subset of samples from the 1000 Genomes Project v3 that had matching ancestry to that study. The second sample set needed raw genotype data and LD was estimated directly for these samples."

18)- p. 13, last paragraph. You chose the causal variant "randomly", but how exactly? Uniformly from the same region/gene, or the entire genome?

To select causal variants we used the sample function in R, corresponding to a uniform draw from the entire chromosome. We have clarified this now in the manuscript (p15).

19)- Table 1., you refer to the APCDR-Uganda data as UG, but do not point this out in the caption, or use UG anywhere else in the paper.

We thank the reviewer for pointing this out. We have corrected this.

REVIEWERS' COMMENTS:

Reviewer #1 (Remarks to the Author):

I appreciate the authors' thorough and thoughtful response to previous concerns raised by this reviewer and others. The authors have made significant improvements in the presentation of their findings and have clarified many aspects of the study design. I have only minor comments/suggestions remaining, largely related to providing clarifications from the response in the main text.

1. Regarding the variants that were selected for inclusion in the final GRS, the response clearly states that you started with the 444 lead variants from the GLGC meta-analysis before selecting independent variants for inclusion. It would be helpful to explicitly state this in the methods as well. Additionally, for the purposes of replicating these findings in the future, a list of variants that were selected in the supplement would be a welcome addition.

2. When the authors state in their response, "For CKB, the HDL GRS had a correlation of $r=0.18$ ($SE=0.02$, $p=1.4 \times 10^{-22}$) and the LDL GRS of $r=0.20$ ($SE=0.02$, $p=32 \times 10^{-26}$) while the triglyceride GRS showed a stronger attenuation relative to UKHLS with $r^2=0.14$ ($SE=0.02$, $p=3.8 \times 10^{-12}$)." They are using r for HDL and LDL and then r^2 for TG. Is this intentional?

3. In some instances, the authors provide clear responses in their rebuttal, but do not address these specifically in the main text. While the rebuttal is published alongside the manuscript, some additional text added to the main text would be beneficial. For example, pointing out that PCs were used as covariates in GRS analyses only for CKB, as other cohorts showed no evidence of inflation, and including lambda GCs, as done in the response. Although, I understand that due to size limitations, all details cannot be included in the main text.

Reviewer #2 (Remarks to the Author):

The authors addressed all questions accordingly. I have no further comments.

Reviewer #3 (Remarks to the Author):

I would like to thank the authors for their thorough response. They have addressed all of my comments and clarified several points. The resulting manuscript is substantially improved, and I recommend that it be accepted for publication.

Minor comments:

- Regarding point 15 in my review, in essence I was suggesting that you try different R2 pruning thresholds for your analysis.

Response to reviewers' comments

Reviewer #1 (Remarks to the Author):

I appreciate the authors' thorough and thoughtful response to previous concerns raised by this reviewer and others. The authors have made significant improvements in the presentation of their findings and have clarified many aspects of the study design. I have only minor comments/suggestions remaining, largely related to providing clarifications from the response in the main text.

1. Regarding the variants that were selected for inclusion in the final GRS, the response clearly states that you started with the 444 lead variants from the GLGC meta-analysis before selecting independent variants for inclusion. It would be helpful to explicitly state this in the methods as well. Additionally, for the purposes of replicating these findings in the future, a list of variants that were selected in the supplement would be a welcome addition. We have now made clear in the manuscript that the 444 loci were the starting point (p14) and have added a supplementary table with the list of loci used for each genetic risk score.

2. When the authors state in their response, "For CKB, the HDL GRS had a correlation of $r=0.18$ ($SE=0.02$, $p=1.4 \times 10^{-22}$) and the LDL GRS of $r=0.20$ ($SE=0.02$, $p=32 \times 10^{-26}$) while the triglyceride GRS showed a stronger attenuation relative to UKHLS with $r^2=0.14$ ($SE=0.02$, $p=3.8 \times 10^{-12}$)." They are using r for HDL and LDL and then r^2 for TG. Is this intentional? We apologise for this typo. We have corrected this. All estimates are correlations (\$r\$ ).

3. In some instances, the authors provide clear responses in their rebuttal, but do not address these specifically in the main text. While the rebuttal is published alongside the manuscript, some additional text added to the main text would be beneficial. For example, pointing out that PCs were used as covariates in GRS analyses only for CKB, as other cohorts showed no evidence of inflation, and including lambda GCs, as done in the response. Although, I understand that due to size limitations, all details cannot be included in the main text.

In the methods it is stated what covariates were included for each study. However, we have now also added the inflation estimates and made it clear why PCs were included for CKB (p12).

Reviewer #2 (Remarks to the Author):

The authors addressed all questions accordingly. I have no further comments.

Reviewer #3 (Remarks to the Author):

I would like to thank the authors for their thorough response. They have addressed all of my comments and clarified several points. The resulting manuscript is substantially improved,

and I recommend that it be accepted for publication.

Minor comments:

- Regarding point 15 in my review, in essence I was suggesting that you try different R2 pruning thresholds for your analysis.

We thank the reviewer for clarifying this and look forward to exploring different ways of estimating genetic risk scores in future research.